# Studying boundary layer methane isotopy and vertical mixing processes at a rewetted peatland site by unmanned aircraft system

Astrid Lampert[1], Falk Pätzold[1], Magnus O. Asmussen[1], Lennart Lobitz[1], Thomas Krüger[1], Thomas Rausch[1], Torsten Sachs[1,2], Christian Wille[2], Denis Sotomayor Zakharov[3], Dominik Gaus[3], Stephan Bansmer[3], and Ellen Damm[4]

[1]TU Braunschweig, Institute of Flight Guidance, Hermann-Blenk-Str. 27, 38108 Braunschweig, Germany
[2]German Research Centre for Geosciences, Telegrafenberg, 14473 Potsdam, Germany
[3]TU Braunschweig, Institute of Fluid Mechanics, Hermann-Blenk-Str. 37, 38108 Braunschweig, Germany
[4]Alfred Wegener Institute, Helmholtz Centre for Polar and Marine Research, Am Handelshafen 12, 27570 Bremerhaven, Germany

**Correspondence:** Astrid Lampert (Astrid.Lampert@tu-braunschweig.de)

**Abstract.** The combination of two well-established methods, of quadrocopter-borne air sampling and methane isotopic analyses, is applied to determine the source process of methane at different altitudes and to study mixing processes. A proof of concept study was performed to demonstrate the capabilities of quadrocopter air sampling for subsequently analysing the methane isotopic composition $\delta^{13}C$ in the laboratory. The advantage of the system compared to classical sampling at ground and at tall towers is the flexibility concerning sampling location, and in particular the flexible choice of sampling altitude, allowing to study layering and mixing of air masses with potentially different spatial origin of air masses and methane. Boundary layer mixing processes and the methane isotopic composition were studied at Polder Zarnekow in Mecklenburg–West Pomerania in the North East of Germany, which has become a strong source of biogenically produced methane after rewetting the drained and degraded peatland. Methane fluxes are measured continuously at the site. They show high emissions from May to September, and a strong diurnal variability. For two case studies on 23 May 2018 and 5 September 2018, vertical profiles of temperature and humidity were recorded up to an altitude of 650 m and 1000 m, respectively, during the morning transition. Air samples were taken at different altitudes and analysed in the laboratory for methane isotopic composition. The values showed a different isotopic composition in the vertical distribution during stable conditions in the morning (delta values of -51.5‰ below the temperature inversion at an altitude of 150 m on 23 May 2018 and at an altitude of 50 m on 5 September 2018, delta values of -50.1‰ above). After the onset of turbulent mixing, the isotopic composition was the same throughout the vertical column with a mean delta value of $-49.9 \pm 0.45$‰. The systematically more negative delta values occurred only as long as the nocturnal temperature inversion was present. During the September study, water samples were analysed as well for methane concentration and isotopic composition in order to provide a link between surface and atmosphere. The water samples reveal high variability on horizontal scales of few 10 m for this particular case. The airborne sampling system and consecutive analysis chain were shown to provide reliable and reproducible results for two samples obtained simultaneously. The method presents a powerful tool for distinguishing the source process of methane at different altitudes. The isotopic composition showed clearly depleted delta values directly above a biological methane source when vertical mixing was hampered by a temperature inver-

sion, and different delta values above, where the air masses originate from a different footprint area. The vertical distribution of methane isotopic composition can serve as tracer for mixing processes of methane within the atmospheric boundary layer.

## 1 Introduction

Methane's ($CH_4$) global warming potential is 32 times that of carbon dioxide ($CO_2$) on a century timescale (Etminan et al., 2016), and 72 times higher on a decadal time scale producing a near-future, greater overall impact on the atmospheric radiative balance (Solomon et al., 2007). As a result, $CH_4$ regulation has a far higher potential for near-term climate change mitigation than $CO_2$. Moreover, global warming feedbacks and rising anthropogenic emissions will likely increase $CH_4$ emissions (Wunch et al., 2009). Yet, current knowledge of $CH_4$ biogeochemical processes, transport and small-scale distribution remains

inadequate. This is in part caused by the complex processes of production, transformation and transport, and in part caused by the lack of sufficiently accurate assessments of the vertical distribution of methane and the temporal and spatial behaviour of highly variable anthropogenic and natural $CH_4$ surface fluxes (Solomon et al., 2007). Global mean $CH_4$ concentration increased until the early 1990s, then mostly stabilized for about a decade (Dlugokencky et al., 2003). Since 2007, growth has resumed (Dlugokencky et al., 2013), which is visible in the remote polar areas as well (Nisbet et al., 2016). The stabilization

was proposed to have been a consequence of reduction in methane emissions from Siberian gas fields and other sources inside the Soviet Union after its collapse, changes in rice agriculture, and changes in microbial emissions (Dlugokencky et al., 2009). The recent increase has been proposed to be related to enhanced emissions from tropical wetlands associated with the extremely wet season in 2009 in the Amazon region (Dlugokencky et al., 2009; Chen et al., 2010).

Isotopic composition allows to distinguish different source categories. Biologically produced methane has a typical $\delta^{13}$C ratio

of -70 to -55‰ (France et al., 2016; Nisbet et al., 2016), methane from fossil fuels has a typical value of -55 to -25‰ (Kirschke et al., 2013), and methane from fires has values of -25 to -13‰ (Kirschke et al., 2013; Nisbet et al., 2016). Further, smaller differences in the isotopic composition are observed for different geographic regions. The background value of air in the free troposphere currently has a value of around -47.4‰ at northern latitudes (Nisbet et al., 2016). The isotopic shift towards depleted (lower, more negative) $\delta^{13}$C values indicates an increasing share of biogenic origin (Nisbet et al., 2016).

The source process of methane and the importance of different natural sources is under discussion for various locations worldwide. Known sources in northern latitudes are permafrost areas (Sachs et al., 2010; France et al., 2016; Sasaki et al., 2016; Kohnert et al., 2018), the Arctic ocean (Yu et al., 2015; Mau et al., 2017) and wetlands (Bellisario et al., 1999). Each source region has a unique isotopic composition, and mixing of different air masses results in a linear combination of the corresponding $\delta^{13}$C values (France et al., 2016). Field measurements of methane isotopic composition have been performed at the Cabauw

tower at a sampling altitude of 20 m, demonstrating the potential of isotopic analyses to determine contributions from isotopically different sources to the dominating source (Röckmann et al., 2016).

However, the classical time-averaging over 30 min is not suitable to capture turbulent small-scale processes for spatially hetero-geneous sources and non-stationary signals, as changes may occur on much smaller temporal and spatial scales. Alternatives are e.g. simultaneous measurements upwind and downwind of heterogeneous methane emitting sources (Denmead et al., 1998). Schaller et al. (2018) showed that methane is frequently emitted from oversaturated ground water or surface water in eposidic outbursts of few minutes duration, which is triggered by atmospheric events. Such events cannot be treated by the commonly applied 30 min averaging.

In the Arctic, inter-annual shifts in the sea ice drift patterns generate an inter-annually patchy methane excess in polar surface water and methane efflux (Damm et al., 2018). The source process of this enhanced methane concentration and the exchange processes between ocean, atmosphere and sea ice are subject to current investigation (Mau et al., 2017; Uhlig and Loose, 2017; Platt et al., 2018). The exchange of gas between air and sea strongly depends on the water stratification (Andersson et al., 2017). Further, isotopic fractionation towards depletion of $\delta^{13}$C in the range of few to several ‰ is observed at the water-air interface for diffusion processes (Happell et al., 1995)

The need to improve understanding of the heterogeneous methane sources and the transition from the surface into the at-mosphere in the Arctic motivated the development of a flexible airborne sampling system, which provides information on atmospheric stability. In this context, unmanned aerial systems (UAS) fill an observational gap for methane mixing processes. They are able to sample small scales with a typical horizontal distance of 1 km, if they are required to be operated in the line of sight, and they reach the top of the atmospheric boundary layer, with a maximum altitude of typically around 1 km. UAS can be operated in remote areas, requiring less infrastructure in comparison with permanent measurement stations, and they can be used more flexibly than manned aircraft, enabling fast reactions to environmental events like changes of emissions through rain, drought, construction, or fire.

First applications of measuring the methane concentration with UAS have been demonstrated: The air sampling inlet inte-grated into multirotor systems is either directly connected to the ground-based methane analyser via a sampling line (Brosy et al., 2017), or the air is stored in a tubing, which is analysed after the flight with a cavity ring down spectrometer (Andersen et al., 2018). The limiting element for both techniques is the length and weight of the sampling line or tube, and sampling altitudes up to 50 m have been published (Brosy et al., 2017; Wolf et al., 2017; Andersen et al., 2018). An air sampling concept based on filling evacuated stainless steel containers by remotely opening a valve and subsequent chemical analyses of trace gases and first applications on multicopter systems have been shown (Chang et al., 2016, 2018). In-situ methane analysers small enough to be integrated on a UAS have been presented (Gurlit et al., 2005; Miftah El Khair et al., 2017; Graf et al., 2018), and first scientific articles showing field measurements of in-situ multicopter borne methane concentrations up to an altitude of 600 m have been published (Golston et al., 2017). The method of air sampling and methane isotopic analyses has been applied to (manned) airborne measurements with high payload capacity in the lower troposphere, e.g. above Siberian peatlands to dis-tinguish emissions from fossil and biogenic emissions (Umezawa et al., 2012), up to balloon borne observations to study sink mechanisms of methane in the stratosphere (Sugawara et al., 1997).

The goal of the study is two-fold:

- Proof-of-concept for the experimental setup of the quadrocopter borne sampling system, and subsequent laboratory analyses, to identify vertical layers of different isotopic composition and therefore the spatial origin and source processes

- Identification of small-scale atmospheric methane inhomogeneities which require the development of new methods for understanding turbulent mixing processes

In order to test the system's capabilities of providing reliable vertical profiles of the isotopic composition, measurements were performed at a rewetted peatland site, Polder Zarnekow (Zerbe et al., 2013), which is known as a source of biologically produced methane (Franz et al., 2016). In the absence of turbulent mixing, which typically happens during night, local emissions of the wetlands produce a depleted delta value compared to the atmospheric background above the temperature inversion. During the morning transition, when the stable stratification is gradually replaced by a convectively mixed atmospheric boundary layer,

the isotopic composition should adjust to a constant delta value throughout the profile within the uncertainties of laboratory air sample isotopic analyses. To support the hypothesis of the small-scale horizontal inhomogeneity at the study site, the methane concentration of water samples from locations within a radius of 100 m was analysed.

## 2   Methods

In the following, the quadrocopter ALICE (Airborne Tool for Methane Isotopic Composition and Polar Meteorological Experi-

ments) as the carrier system, the payload consisting of the air sampling subsystem and the meteorological sensors and data acquisition are described. Further, the laboratory air analysis procedures, and the measurement site for system tests are introduced. For evaluating the whole measurement chain, a local source of methane of particular isotopic composition, and atmospheric conditions that first inhibit and then enforce vertical mixing processes (morning transition) were required. These conditions were met at Polder Zarnekow on 23 May 2018. For confirming the results, the same measurement strategy was

applied to the same site on 5 September 2018.

### 2.1   Quadrocopter ALICE and instrumentation

The quadrocopter ALICE was designed as platform to carry meteorological sensors and 12 glass bottles for air sampling (Fig. 1). The construction of the quadrocopter was calculated for the specific tasks and the payload described in the following. Therefore, all relevant load cases that were expected during the flight were applied in analytical and numerical models to op-

timize the structure of the quadrocopter. Modern manufacturing methods like selective laser sintering and laser cutting where used to build the structure as light-weight as possible but as stable as necessary. ALICE has dimensions of 1.82 m x 1.82 m x 0.78 m including the scientific payload. The arms are quickly removable (rotating lock) for convenient storage and transport. At a tare weight of 6.3 kg, ALICE's maximum take-off weight is 25 kg. For the operations presented here, the total weight was 19 kg, which is composed of 6.3 kg the quadrocopter system itself, 4.9 kg of LiPo batteries with a total capacity of 21 Ah

and a nominal voltage of 44.4 V for rotor power supply, and 7.8 kg payload including sensors, glass bottles, data acquisition, power supply for payload and a safety parachute of 12 m$^2$. The payload is placed in the center of the system on a platform with

dimensions of 370 mm x 370 mm. The temperature and humidity sensors are located in a housing to shield against radiation and protect against impact of dust at the edge of the platform. An electronics box contains the central data acquisition.

The quadrocopter is constructed with a thrust-to-weight ratio of 2:1. Each propeller has a diameter of 330 mm. The system was designed for wind speeds up to 35 km h$^{-1}$ during take-off and landing, and up to 70 km h$^{-1}$ during free flight. At a wind speed of 60 km h$^{-1}$, a climb rate of the current system of 8 m s$^{-1}$ was still possible. ALICE has been designed to reach altitudes up to 1 km. It is electrically powered with four motors U11 KV120 of T-Motor, China.

As the system was intended for operations in the polar regions, the design point of the system is -30 °C. All parts have been tested extensively in a climate chamber down to -30 °C, taking into account vibrations of the carrier, which were simulated with a shaker after in-flight measurements. The pre-heated batteries are insulated passively, and the temperatures of the batteries and the internal avionics are monitored.

Air temperature is recorded with various temperature sensors of different behaviour. Fast fine wire temperature sensors, manufactured at the Institute of Flight Guidance (Bärfuss et al., 2018), have the advantage of a high temporal resolution up to 100 Hz. Further, standard long-term stable sensors were used (Pt1000 "Humicap" HMP110, Vaisala, Finland, digital sensor TSYS01, Measurement Specialities, US). Relative humidity is measured with two different sensors, the Humicap HMP110 (Vaisala, Finland), and the Rapid P14 (Innovative Sensor Technology, Switzerland). Absolute pressure is recorded with AM-SYS 5812-0150-B sensors, AMSYS, Germany. Two pyranometers ML-01 of EKO Instruments, Japan, with nadir and zenith viewing geometry are integrated, which allow to estimate cloudiness and surface properties at a sampling rate of 100 Hz. Further, a surface temperature sensor Melexis MLX90614, Belgium, is mounted into the airframe and fixed at the bottom of the UAS. A Global Navigation Satellite System (GNSS) receiver and an inertial measurement unit (IMU) ADIS16488 (Analog Devices, US) are integrated. Data is recorded at a sampling frequency of 100 Hz.

The air sampling system consists of 12 glass flasks (sample containers) of 100 ml volume, which are evacuated before take-off. Their arrangement with respect to the copter can be seen in Fig. 1. The distance from the tip of the closest rotor blades to the inlet is approximately 7 cm. The glass flasks are equipped with two manual valves, one on each side, and additionally one electromagnetic valve, which is applied only during the flight (Fig. 2). Directly before the mission, each glass flask is linked with a vacuum pump RE5 of Vacuubrand, Germany. One valve is left open, and an electromagnetic valve is connected, which is normally closed. Then the flask is evacuated, and the pressure is controlled by a pressure sensor integrated in the electromagnetic valve. The flasks are opened during the flight with magnet valves, that are triggered either manually by remote control or automatically at altitudes predefined by the operator. After triggering, ambient pressure is reached within approximately 1.3 s. The pressure sensors integrated in the valves are used to monitor air tightness. The most delicate component are the manual plastic valves, used to close the glass flasks for transport, which are designed to be air tight when closed, but not when open. They had to be treated individually and controlled to make sure that no leakage occurred during the mission. For quality control and redundance, two glass flasks were filled simultaneously, resulting in six possible sampling altitudes during one flight.

The onboard data are downlinked to a ground station and displayed to the operator. Depending on the atmospheric structure, the operator decides on the altitude of taking samples during the descent, e.g. above/below the temperature inversion, or within altitudes of enhanced humidity, as required for the scientific question. The whole mission can be flown automatically by a

PixHawk autopilot. It is supervised by a safety pilot and a scientific operator who is controlling the system and performance as well as the measurements. Two small cameras, one pointing downwards (GoPro HERO5 Black, 12 Megapixel), one looking to the side (GoPro HERO Session Actionkamera, 8 Megapixel), were integrated. The captured video of the downward pointing camera was transmitted to the operator with 720p resolution and 60 Hz. There are different telemetry connections: A 2.4 GHz link is used for the remote control. A 868 MHz connection serves to send commands to the autopilot, and a 868 MHz link is used for activating the safety parachute trigger. Further, a 5.8 GHz video link is established. Scientific data are transmitted via a 433 MHz connection.

## 2.2 Quadrocopter flow simulations and impact on sampling

In order to quantify the effect of the vertical flow induced by the quadrocpter, numerical simulations were performed with the software ANSYS CFX. The simulations were transient in nature using a Reynolds-Averaged Navier Stokes (RANS) approach with the Shear Stress Transport (SST) turbulence model (Menter, 1994). A simplified model of the propeller blade was used, with a multidomain approach: The blade is enclosed in a rotating domain, surrounded by a static domain. Simulations were performed for hover with a propeller rotation speed of $3167 \, min^{-1}$, for vertical climb at a speed of $6.5 \, m \, s^{-1}$ with a rotation speed of $3913^{-1}$ and of vertical descent at a speed of $-2.5 \, m \, s^{-1}$ with a rotation speed of $2880 \, min^{-1}$. An ambient temperature of $0°$ C and pressure of 1023 hPa were considered. Contours of relative vertical velocity show a core region of positive relative velocity directly below the center of the blade, and a negative relative velocity up to $19 \, m \, s^{-1}$ below the blade for a distance exceeding 0.75 m (Fig. 3). Additionally, zones of recirculation can be seen around the tips of the propeller, especially for the descent case. The air sampling system is contained in the middle of the copter, and is less affected by artificial turbulence than the areas below the rotor blades.

Assuming that in the worst case the sampling takes place within the downwash of the rotor blades of not more than $-19 \, m \, s^{-1}$, the sampling time of of 1.3 s duration results in a vertical resolution of around 25 m. Sampling during descent with a speed of $-2.5 \, m \, s^{-1}$ adds an uncertainty in the altitude of 5 m. Altogether, the sampling is influenced by air in a height interval of 30 m. This is the sufficient for the sampling intervals of around 100 m, and for determining that the sampling was done below or above the temperature inversion.

A more realistic estimate of the altitude interval uncertainty assumes the vertical velocities directly next to the rotor blade tips (See Fig. 1): The highest induced speed in close proximity of the tips of the rotor blades during descent is $-8 \, m \, s^{-1}$ (Fig. 3). This results in an altitude interval of around 10 m caused by the flow field plus an altitude interval of 5 m due to the vertical descent speed. Therefore, a realistic estimate of the altitude influencing the sampled air is 15 m.

## 2.3 Laboratory isotopic analyses

Following the quadrocopter mission, the sample containers (SC) were transported to the laboratory at the Alfred Wegener Institute in Bremerhaven, Germany for analysis of the isotopic composition. The $\delta^{13}C$ value of the air samples was analyzed using a Delta plus XP mass spectrometer combined with a combustion oven, a gas pressure interface and a pre-concentration

device (PreCon) (ThermoFinnigan, Bremen, Germany). All valves and traps are automatically operated by the mass spectrometer software. Each SC was installed onto the PreCon and the connections were flushed with helium before the SC was opened. The air sample was first carried by helium carrier gas through a chemical trap and then trapped in a cool box filled with liquid nitrogen (-196°C) to remove $CO_2$, CO and $H_2O$. Afterwards the $CO_2$ free air was carried by helium carrier gas to the combustion oven (1000°C) for methane conversion into $CO_2$ which then was purged into a second cool box filled with liquid nitrogen and trapped therein. When the combustion is finished, the gas stream is purged and trapped in a third cool box filled with liquid nitrogen to pre-concentrate the sample, and carried by helium carrier gas into the isotope ratio mass spectrometer. Methane stable carbon isotope data are given in $\delta$ notation (in ‰) relative to the Pee Dee Belemnite (PDB) standard:

$$\delta^{13}C\text{‰} = \left( \frac{\frac{^{13}C}{^{12}C} sample}{\frac{^{13}C}{^{12}C} reference} - 1 \right) \cdot 1000 \tag{1}$$

Precision, determined as the maximum difference of delta values during repeated analysis of ambient Bremerhaven air samples taken simultaneously and analysed consecutively many times per year, is better than $0.5$‰. Further applications of the system have been described in Mau et al. (2013); Damm et al. (2015); Verdugo et al. (2016); Fenwick et al. (2017).

## 2.4 Study site

The shallow Polder Zarnekow with a water depth of less than 1 m belongs to a large area of rewetted peatlands in the Peene River valley in North-Eastern Germany (53°52.5' N 12°53.3' E, less than 0.5 m a.s.l). It formed after the dikes were opened in 2004/05 in order to restore the peatlands, taking up $CO_2$ to reduce greenhouse gases. The total rewetted area is 421 ha in size (Gelbrecht et al., 2008). The site is a Fluxnet site (DE-Zrk) and is part of the Northeast German Lowland observatory of the Terrestrial Environmental Observatories Network TERENO (Heinrich et al., 2018).

It is equipped with state of the art eddy covariance (EC) instrumentation recording the wind vector, temperature and the concentration of water vapour, $CH_4$ and $CO_2$ at a frequency of 20 Hz. The measurement height above the water surface is around 2.6 m, depending on the water level. Water vapour and $CO_2$ are measured with a LI-7200 sensor of LI-COR, US. Methane is measured with a LI-COR sensor LI-7700. Further, a Los Gatos Fast Greenhouse Gas Analyser (FGGA, US) records the concentration of the three greenhouse gases at a frequency of 20 Hz. Four automatic measurement chambers are installed along a transect between the EC system and the shore of the lake for spatially resolved $CO_2$ and $CH_4$ flux investigations (Hoffmann et al., 2017).

The restoration of the peatland area towards a net sink of the greenhouse gas $CO_2$ is a process of several years to decades. Initially, the restoration is accompanied by a strong increase in $CH_4$ emissions, which depend on vegetation and the water level (Couwenberg et al., 2011; Zak et al., 2015). The shallow eutrophic lake in particular acts as a strong source of $CH_4$ (Franz et al., 2016). Maximum methane emissions are typically observed in summer (Franz et al., 2016). In the diurnal cycle, the maximum $CH_4$ emissions were recorded during night (Franz et al., 2016). This is in agreement with stronger convective mixing of the lake induced thermally from May to October (Franz et al., 2016), which leads to diffusive $CH_4$ emissions (Hoffmann et al., 2017). Time series of methane fluxes of the 3-year period 2016 to 2018 reveal a high variability over the entire period and

within time periods of few days, with noticeable fluxes during the growing season between May and September (not shown).

## 2.5 Water sampling

For identifying the reason for small-scale inhomogeneities of the atmospheric methane isotopic composition, the methane source located within the surface water was sampled at different locations on 5 September 2018. Six water samples were taken with Kemmerer glass bottles of 50 ml at the locations Z-1 to Z-6 indicated in Fig. 4. After filling and locking with screw cap, the water samples were stored and transported light-tight in a cool box. They were analysed in the laboratory for methane concentration and isotopic composition at the Alfred Wegener Institute in Bremerhaven on 6 and 7 September 2018, thus directly after the sampling.

## 2.6 Flight strategy

The aerial measurement strategy consists of an automatic climb flight up to 1 km altitude and down again with real-time data transmission of selected parameters at 1 Hz. For the missions presented here, a permission from the nature protection agency and coordination with the German Civil Aviation Agency (DFS, Deutsche Flugsicherung) were required. Flights were permitted between sunrise and sunset. As it takes some hours until the nocturnal temperature inversion is replaced by a well-mixed boundary layer with increasing solar radiation, it was possible to take the air samples during the transition from nocturnal stable boundary layer to the convectively mixed boundary layer.

On 23 May 2018, three consecutive measurement flights of around 10-11 min duration were performed over a time period of 3 h in the morning, with take-off times 06:04, 07:30, and 08:29 and UTC, corresponding to a local time between 08:04 and 10:29. Sunrise was at 04:55 local time.

Each flight consisted of manual take-off and climb up to around 50 m altitude, then handing over the system to the autopilot. The flight pattern followed 3 waypoints at 50 m altitude leading to a position directly above the open water fraction of the polder. There, a vertical ascent with a mean vertical speed of $5\,\mathrm{m\,s^{-1}}$ up to a height of 650 m agl and subsequent descent with a vertical speed of $-2\,\mathrm{m\,s^{-1}}$ took place. For this feasibility study, air samples were taken during descent at the approximate altitudes of 600 m, 400 m, 300 m, 200 m, 100 m and 10 m. Then the quadrocopter flew to a waypoint close to the landing point and was landed manually. During ascent, the temperature profiles were studied, as a base to plan sampling altitudes for the descent. The air samples were analysed in the laboratory for methane isotopic composition at the Alfred Wegener Institute in Bremerhaven on 11, 12 and 13 June 2018, thus 3 weeks after the sampling.

On 5 September 2018, five consecutive measurement flights of around 12-13 min duration were performed over a time period of 5 h in the morning, with take-off times 06:04, 07:15, 09:12, 10:05 and 10:57 UTC, corresponding to a local time between 08:04 and 12:57. Sunrise was at 06:23 local time. The same flight strategy as described above was applied. The ascents reached an altitude of 1000 m, but the sampling altitudes remained the same. The vertical ascents were done with a mean vertical speed of $6.5\,\mathrm{m\,s^{-1}}$ and the descents with a vertical speed of $-2.5\,\mathrm{m\,s^{-1}}$.

# 3 Results

## 3.1 Synoptic conditions and meteorological observations

On 23 May 2018, the synoptic situation was characterized by a pronounced high-pressure system above Scandinavia and the Baltic Sea, leading to conditions of low wind speed below $5\,\mathrm{m\,s^{-1}}$, north-easterly wind directions and a cloudless sky, as indicated from the smooth behaviour of global radiation (Fig. 5).

On 5 September 2018, the synoptic situation was determined by two strong high pressure systems located above the Atlantic Ocean and above Northern Russia, and a low-pressure system above Southern Europe. This resulted in low wind speed below $4\,\mathrm{m\,s^{-1}}$ of north-easterly direction at the observation site, similar to the conditions on 23 May 2018. As the eddy flux tower is located at the North-East of the polder, flux footprints are influenced by areas outside of the polder area (see Figs. 11 and 4). As the profile measurements were performed directly above the polder, the sampled air has a different footprint. The locations of the water samples are all outside the footprint area of the EC tower.

In the morning, until 05:30 UTC, around 30 min before the first flight, fog was observed, which was denser towards the East. Starting around 09:00, before flight 3, shallow convective cumulus clouds were present. This is also evident in the high variability of the global radiation (Fig. 6).

On 23 May 2018, the near-surface temperature increased from around 13-14°C at 6 UTC to around 17°C at 08:30 UTC (Fig. 7). The profiles of potential temperature show a strongly stable stratification in the morning, with an increase of potential temperature of around 2°C (06:04 UTC) from the surface to 250 m. The top of the nocturnal temperature inversion was located at around 150 to 250 m altitude. At this altitude range, the water vapour mixing ratio decreased as well from around $7\,\mathrm{g\,kg^{-1}}$ in the boundary layer close to the surface to around $5\,\mathrm{g\,kg^{-1}}$ above. After 07:30 UTC, the atmosphere was less stably stratified. The water vapour mixing ratio was enhanced in the lowermost 150 m at 06:04 UTC. Above and later, the water vapour mixing ratio was almost constant with height.

On 5 September 2018 during the first flight, the profiles show a mixed-layer up to 50 m altitude and a stable stratification above (Fig. 8). The water vapour mixing ratio was lower in the mixed layer ($11\,\mathrm{g\,kg^{-1}}$) and increased above up to $14\,\mathrm{g\,kg^{-1}}$. For the following four flights, the near surface temperature increased, and the potential temperature increased only slightly with altitude. The water vapour mixing ratio decreased with time throughout the whole profile.

## 3.2 Observations of methane concentrations and isotopic composition

For both observation days, the methane concentration recorded at the eddy covariance tower at 2.6 m altitude above the surface shows a high variability during the night and the morning, and a relatively constant concentration during the day. During the time period of the first flight on 23 May 2018, the methane concentration was still slightly enhanced compared to the background value during the day (Fig. 5). The first flight on 5 September 2018 took place during the high fluctuations of the $CH_4$ concentration (Fig. 6).

The isotopic composition of the air samples was different during the first flight of each day at the altitudes located within the

stable stratification. On 23 May 2018, the delta value was depleted at 10 m and 100 m altitude with a value of -51.5‰ (see Fig. 9). Above, the mean delta was -50.1‰, almost the same as during the next flights throughout the profile, with an average value of -49.9‰. On 5 September 2018, the same systematic behaviour was observed: During the first flight, the delta value was depleted at an altitude of 10 m, in agreement with a temperature inversion above an altitude of 50 m. Above, the delta value of

5    -49.9‰ was the same as during all other flights throughout the profile (Fig. 10).

Aerial pictures obtained during the measurement flights for the two case studies show the difference in water level: On 23 May 2018, the lake was filled with water (Fig. 11). On 5 September 2018, almost the whole lake had fallen dry, with only few wet areas remaining, and the sediment still saturated with water (Fig. 4). The depth was in the range of few cm.

The water samples taken on 5 September 2018 within a radius of 100 m revealed highly different $CH_4$ concentrations (see

Table 1). The highest $CH_4$ concentration of around 4770 ppm was measured from the water sample Z-4 in one of the small remaining water areas with a depth less than 5 cm within the former polder. Directly next to this, the water sample Z-3 taken at a depth of 5-10 cm had a methane concentration of 2310 ppm (see Figs. 4 for exact locations). The water sample Z-4 contained more suspended sediment load than sample Z-3. The sample sites are located within small remaining water areas, which are not connected.

**4   Discussion**

**4.1   Plausibility of the observed isotopic composition**

Methane flux data indicate that the observation site was a source of methane during both measurement days, but had much higher emissions on 23 May 2018 (not shown). During the first flights in the early morning, under stable stratification, an enhanced methane concentration representing local emissions close to the observation site can be expected (Röckmann et al.,

2016), as observed by Andersen et al. (2018) above wetlands. A stable stratification was present at the beginning on both measurement days with the UAS. An enhanced methane concentration and high variability was observed until around 06:30 UTC on 23 May 2018 (Fig. 5), and even higher concentrations and stronger fluctuations were observed until 6:45 UTC on 5 September 2018 (Fig. 6). Under stable atmospheric conditions and low wind speed smaller than $2\,\mathrm{m\,s^{-1}}$ during the night and in the early morning hours, the near-surface methane concentration is enhanced, as vertical mixing hampered, in agreement

with Emeis (2008); Brosy et al. (2017). This can be seen in the time series of the methane concentrations for both days as well (Figs. 5 and 6). During the flights, the methane concentration was already smaller again, which can be explained by vertical mixing up to the temperature inversion. This mixing also influences the isotopic composition. With a local methane source, the near-surface methane concentration above agricultural land therefore increases during stable atmospheric conditions (Wolf et al., 2017), as this prevents mixing with the free troposphere. However, a high local horizontal inhomogeneity in methane

concentrations resulting from mixed-use areas has been reported, an effect which is visible in the lowermost 10 m above the surface, but smeared out above due to an increase in horizontal wind speed (Wolf et al., 2017). In the presented case, the parallel samples obtained at 10 m altitude seem to be mostly of the same spatial origin with small differences in the delta value. Above, differences in the delta values are higher. A wind speed of less than $2\,\mathrm{m\,s^{-1}}$ during the sampling results in horizontal sample

integration over not more than 4 m at the sampling altitude. Small-scale differences in methane isotopic composition can be introduced by mixing processes, and may be reinforced by mixing induced by the quadrocopter system. A high variability of methane sources is in agreement with the highly variable methane water concentrations measured within a radius of 100 m on 5 September 2018.

5  The difference in delta values obtained for air samples near ground and above the temperature inversion during stable stratification is around 1.5‰, thus significantly higher than the uncertainties (flight 1 for each measurement day). This shows that the observed systematic differences are not caused by measurement uncertainty, but by small-scale inhomogeneities and turbulent mixing processes. Other methane sources in the surroundings of the polder are presumably biological source processes as well, they may include larger areas of the rewetted peatland and ruminants, with similar isotopic composition (Röckmann et al., 10  2016).

Assuming that the parts of the surface with high methane concentrations, like sample Z-3, Z-4 and Z-6, act as a methane source, with a delta value of around -48 to -49‰ (water sample Z-4 and Z-6), an isotopic fractionation of around 3‰ would occur across the water-air boundary. This is in the order of magnitude of carbon isotope depletion observed for emitted relative to floodwater $CH_4$, ranging from 1.8 to 3.4‰ (Happell et al., 1995), thus can be considered realistic.

15  **4.2   Interpretation of isotopic composition from measurement point of view**

The isotopic composition of the two air samples taken on 5 September 2018 simultaneously but with a constant horizontal distance of 13 cm agree within 0.1‰ at the lowest altitude of 10 m for Flight 1 to 4 (Fig. 10). This is a much better agreement than the uncertainty of 0.5‰. However, this value has been determined experimentally for long laboratory time series, and may be much better for subsequent analyses. Besides this strong locally and temporally related agreement of the isotopic 20  composition, for other altitudes and flights this difference is in the range of the uncertainty. A possible explanation for the systematic differences is a combination of spatial inhomogeneities and turbulent mixing processes. There are several striking patterns in the profiles of isotopic composition, from lower to higher altitudes:

  – On 5 September 2018 the difference in delta values between the two simultaneous samples is systematically smaller at 10 m altitude (much better than the uncertainty) compared to the higher altitudes, except the last profile.

25  – On 5 September 2018 the differences in delta values at 100 m altitude increase during the course of the day. The profiles of differences in delta values exhibit similarities for parts of the profile between subsequent flights on both days.

  – The delta values are significantly more negative in the morning before vertical mixing starts, as long as a temperature inversion is present (first flight on 23 May 2018 below 150 m, and first flight on 5 September 2018 below 70 m). The difference between delta values below and above the temperature inversion is larger than the uncertainty. This is in 30  agreement with methane from biologic processes emitted from the surface that are not vertically mixed.

Although the differences of the delta values only slightly exceed the uncertainty, there are indications that the differences in delta values are physically present in the air samples: The order of analysing the air samples was chosen randomly and the differences in delta values show a similar height profile for subsequent flights. Two aspects can be highlighted:

- An ideally vertically stratified delta value would not be sampled by the present system as the very dynamic circulation process around the quadrocopter does not result in a homogenously mixed air at the sample ports on the quadrocopter. On the contrary, this circulation process can even amplify natural inhomogeneity. It is assumed that the differences in delta values indicate natural inhomogeneity caused by turbulent mixing processes, but it is not possible to prove it based on the data set.

- Besides the vertical turbulent mixing, the natural spatial inhomogeneity of delta values is not known for the measurement site. Small-scale horizontal variability can be induced by inhomogeneous sources. Episodic $CH_4$ outbursts on short time scales of few min have been observed by Schaller et al. (2018). The high spatial and temporal variability of methane concentration and isotopic composition reported here is in agreement with their observations. Such variability of methane emissions at the field site as well as of the potential upwind $CH_4$ sources may cause the inhomogeneous character of the air samples.

Respecting that the air sample profiling gives a snapshot of a turbulent mixing process, a clear transition in the vertical distribution of the delta values can be seen.

## 4.3 Improvement potential for multicopter based air sampling

According to the simulations, undisturbed air sampling with the multicopter is possible with a sideways pointing inlet that reaches 25 cm beyond the rotor or a tube that reaches 50 cm above the rotors for hover and climb. Air sampling during descent experiences more additional disturbance by the rotor blades and therefore should be compared with air sampling during climb. The initial operation idea was to observe at first the atmospheric stratification in climb and determine the sampling altitudes for subsequent descent based on the altitude of the temperature inversion. However, the first simulation results quantify the difference in additional vertical velocities induced by the measurement system, which are much higher during descent (Fig. 3). Due to efficiency reasons, the vertical climb speed is higher than the descent speed for the current ALICE system. The impact of the climb speed has to be taken into account for the temporal resolution of the sensors. In order to closer constrain the altitude interval of the sampled air, measurements during hover or slow climb flight in combination with an inlet tube of the dimensions mentioned above would be preferable for continuous sampling. However, for the presented sampling system with small volume, the air volume contained in the tubes is not exchanged continuously and would further induce uncertainties. Sampling during slow ascent or hover requires adjusting the battery capacities or the flight mission, e.g. the maximum flight altitude. Further, simulations of the whole multicopter system including the payload are required to quantify the flow field and find the optimal sensor location.

For a systematic comparison with the eddy flux measurements, temporally and spatially integrated measurements would be adequate. However, averaging is only suitable for sufficiently homogeneous surface conditions and emissions. This is not the case here, as already indicated by the different methane concentrations and isotopic compositions for the water samples. Therefore, the instantaneous point samples cannot be compared directly with the classical micrometeorological methods like 30 min averaged EC analyses. For further investigating small-scale inhomogeneities, new methods for observations and analyses are

required: Instantaneous profiles of the methane concentration and isotopic composition upwind and downwind of an EC tower could be combined with wavelet analyses instead of EC covariance analyses, as suggested by Schaller et al. (2018).

## 5 Conclusions

The measurements serve both as a proof of concept for the system and show the vertical mixing of methane in the ABL by means of its isotopic composition. With ALICE air samples and subsequent laboratory analyses, it is possible to determine differences in the methane isotopic composition caused by atmospheric stability.

In summary, the first application of ALICE and the analyses of the air samples show potential for improvement for future missions:

– Air sampling during climb and hover is much less influenced by rotor induced turbulence than during descent.

– Double sampling is highly recommended for system assessment.

– The punctual air sampling for delta value determination should be complemented with simultaneous onboard fast and accurate methane concentration measurements. Light-weight instruments with sufficient accuracy and temporal resolution might be operational in the near future.

The differences in delta values of water and air, the differences in delta values between both flight days and the development during each day emphasize the highly complex and inhomogeneous nature of methane processes on horizontal scales below 1 km in sediments, at the sediment-water and the water-atmosphere interface. Therefore, a suitable method is required for quantifying small-scale inhomogeneous methane sources. Vertical layering of air masses with different methane properties strongly depends on atmospheric stability, both concerning concentration as well as the isotopic composition. A holistic approach is needed to investigate methane processes from sediments to the atmospheric boundary layer, including dedicated measurements of the isotopic fractionation. Despite some points that can be improved, the first applications of ALICE for air sampling and methane isotopic analyses show the potential to contribute substantially to investigate layering and mixing processes of atmospheric methane of different sources. The use of the multicopter represents an advantage over air sampling at tall towers, as it is much more flexible and easier to apply.

*Data availability.* The data of the flight is available upon request from the authors of TU Braunschweig. Biomet and flux data will be uploaded to the European Fluxes Database Cluster (http://www.europe-fluxdata.eu/) and the TERENO Data Portal (http://teodoor.icg.kfa-juelich.de/ddp/) after final processing and quality control.

*Author contributions.* AL wrote the paper, FP developed the quadrocopter payload, TK developed the quadrocopter, FP, TK, TR, AL and MA conducted the measurement campaigns, FP and ED performed the laboratory methane isotope analyses, CW and TS performed the

methane flux measurements, LL performed the quality check of the copter measurements, DG, DSZ and SB performed the simulations. All authors contributed to and commented on the manuscript.

*Competing interests.* The authors declare that they have no conflict of interest.

*Acknowledgements.* This work was supported by the Deutsche Forschungsgemeinschaft (DFG) in the framework of the priority programme
5  "Antarctic Research with comparative investigations in Arctic ice areas" by grants LA 2907/8-1 and DA 1569/1-2. The authors would like to thank Barbara Altstädter and two anonymous referees for critically reading the manuscript.

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

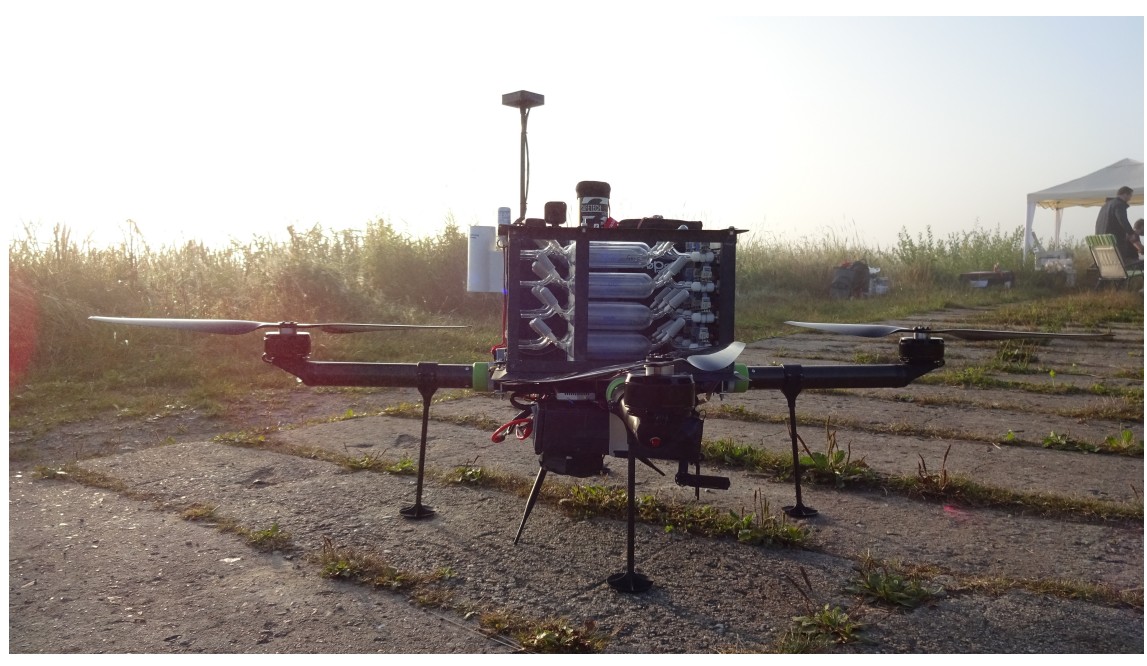

**Figure 1.** The quadrocopter ALICE before take-off in Zarnekow on 5 September 2018.

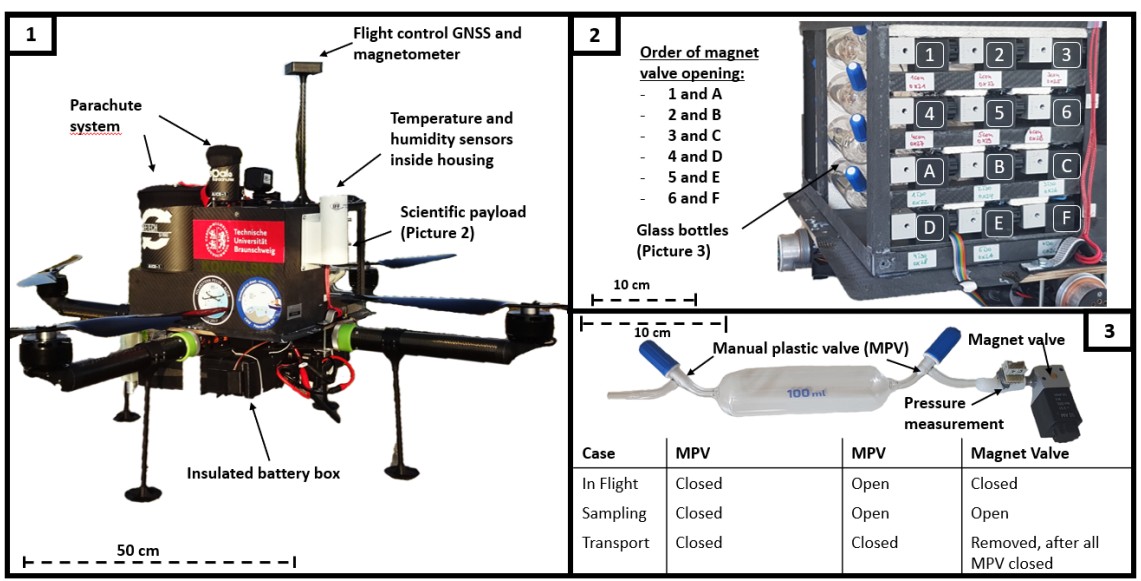

**Figure 2.** ALICE vital components. 1: Overall view of the system 2 : Gas sampling payload consisting of 12 evacuated glass flasks which can be filled by opening an electromagnetic valve during the flight. The positions of the two samples taken simultaneously are indicated. 3: Principle of air sampling with manual and electromagnetic valves, showing the configurations of the valves during flight, sampling and transport.

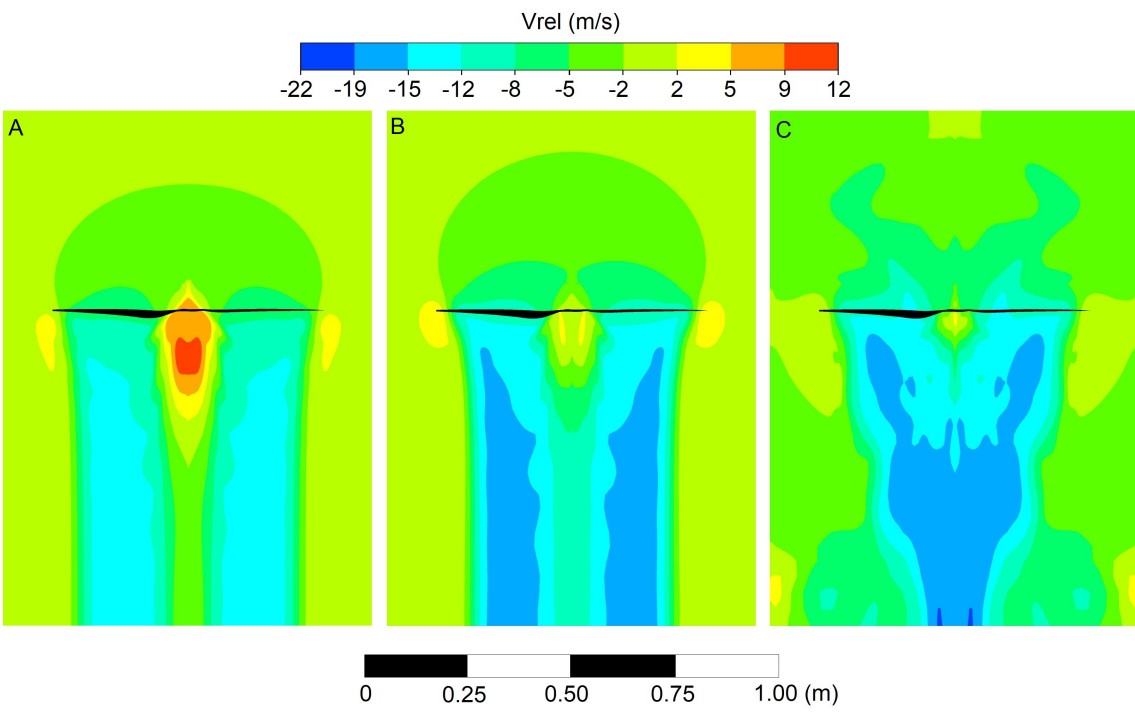

**Figure 3.** Simulation of the flow induced by a propeller blade during hover (A), vertical climb at a speed of 6.5 m s$^{-1}$ (B) and descent at a speed of -2.5 m s$^{-1}$ (C). Downward pointing velocities have a negative sign.

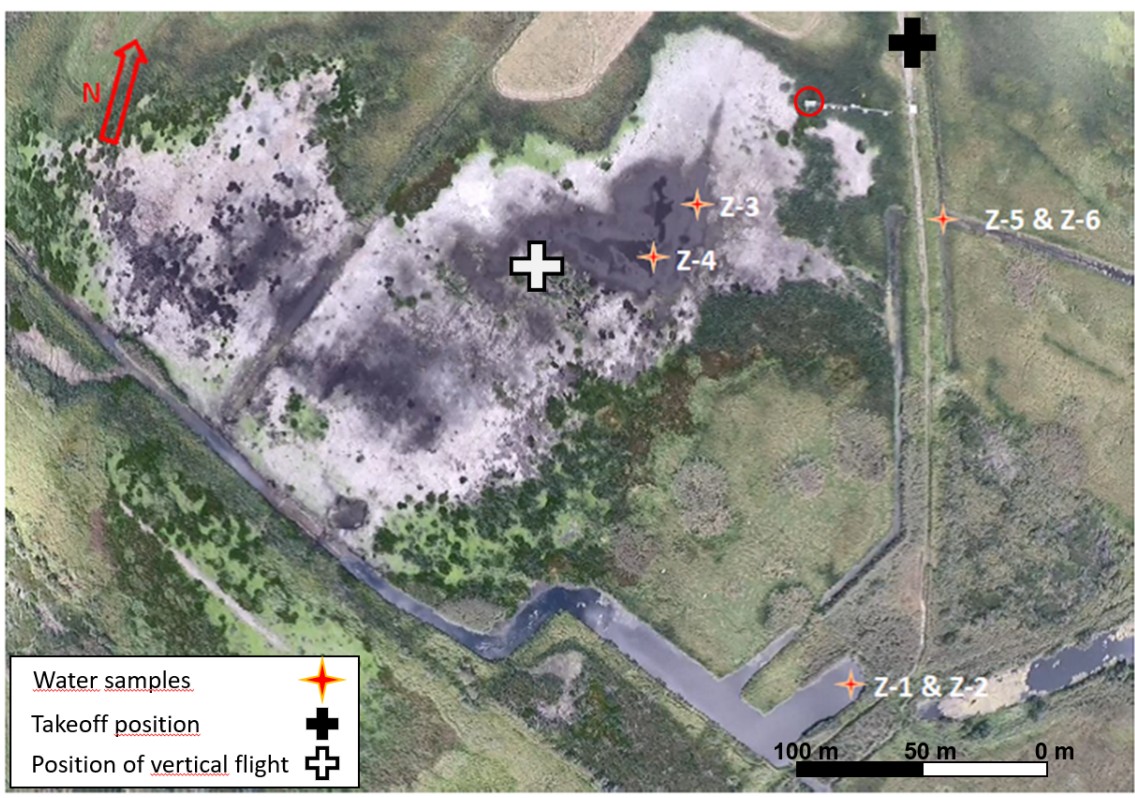

**Figure 4.** Aerial picture of the polder Zarnekow obtained with the quadrocopter ALICE on 5 September 2018. Almost the whole polder fell dry after the extremely warm and dry summer 2018. The sites where water samples were taken are indicated with Z1 to Z6. The location of the EC station is indicated with a red circle.

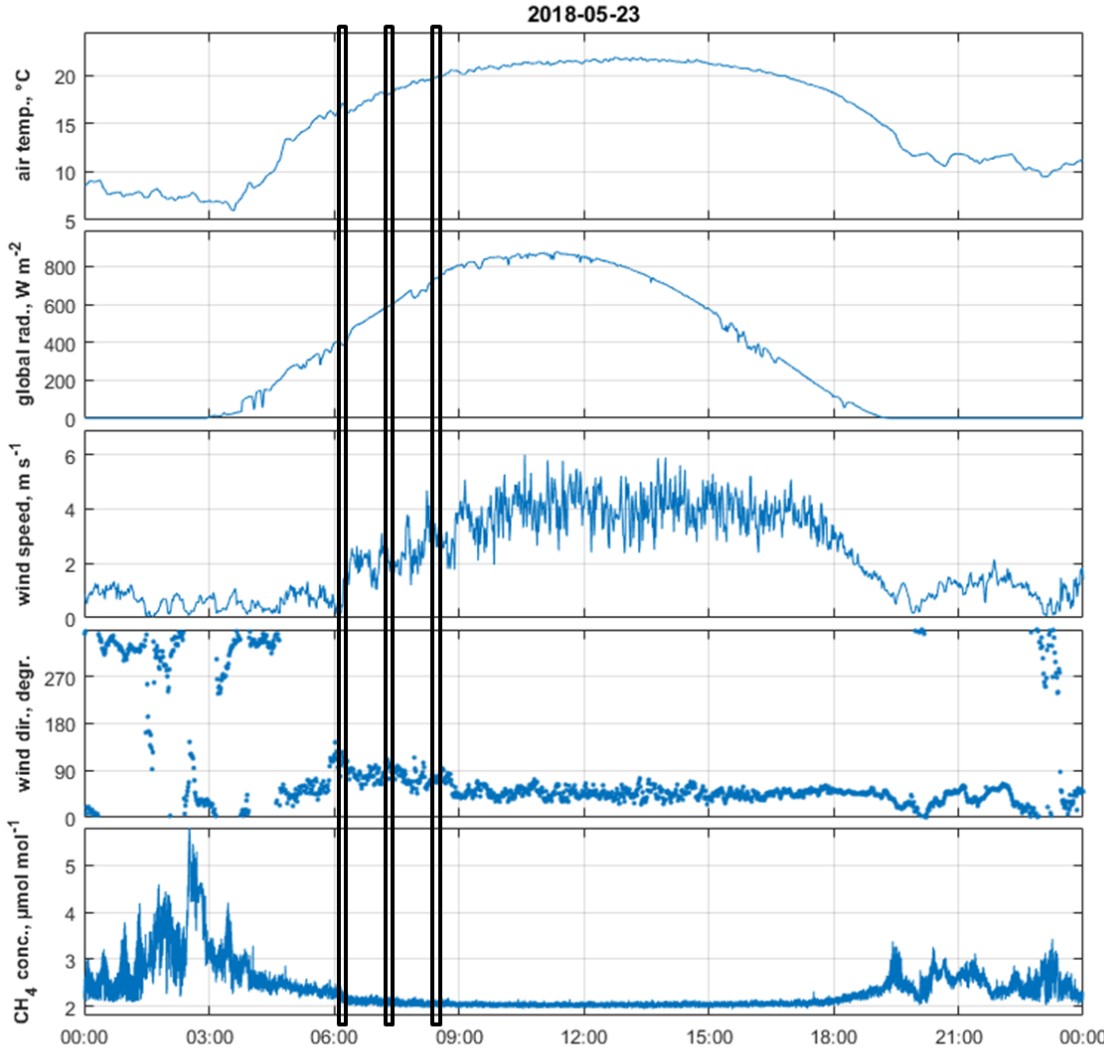

**Figure 5.** Diurnal course of the main meteorological parameters air temperature, global radiation, wind speed, wind direction, and methane concentration (closed-path Los Gatos sensor) recorded at the meteorological mast at Zarnekow on 23 May 2018. The times of the quadro-copter air sampling are indicated by vertical boxes.

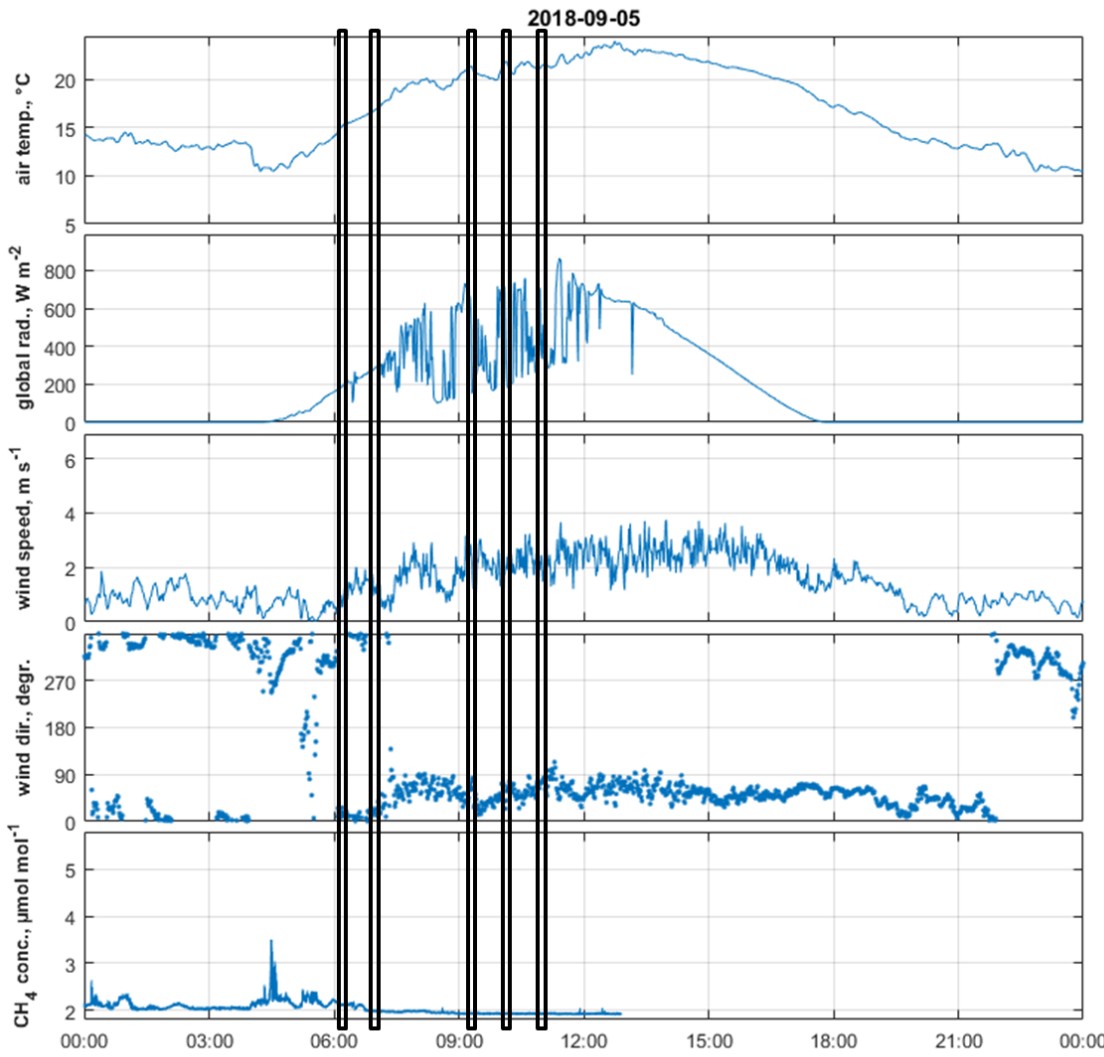

**Figure 6.** Diurnal course of the main meteorological parameters air temperature, global radiation, wind speed, wind direction, and methane concentration (closed-path Los Gatos sensor) recorded at the meteorological mast at Zarnekow on 05 September 2018. The times of the quadrocopter air sampling are indicated by vertical boxes.

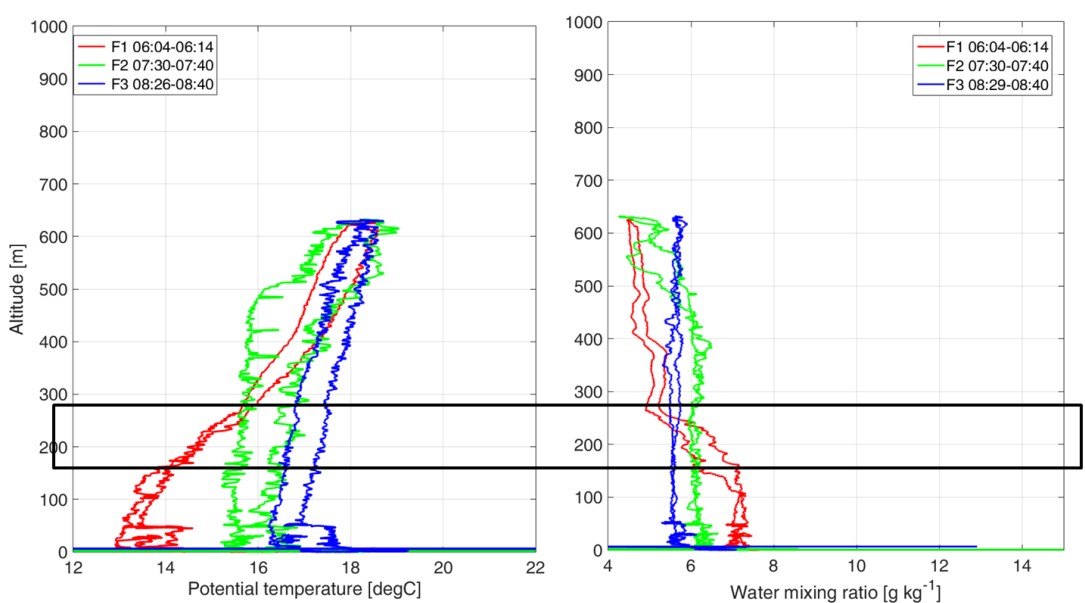

**Figure 7.** Profiles of potential temperature and water vapour mixing ratio obtained on 23 May 2018. The times of the five flights are given in UTC. The horizontal box represents the height interval of the temperature inversion, which is also visible in the large changes of the water vapour mixing ratio.

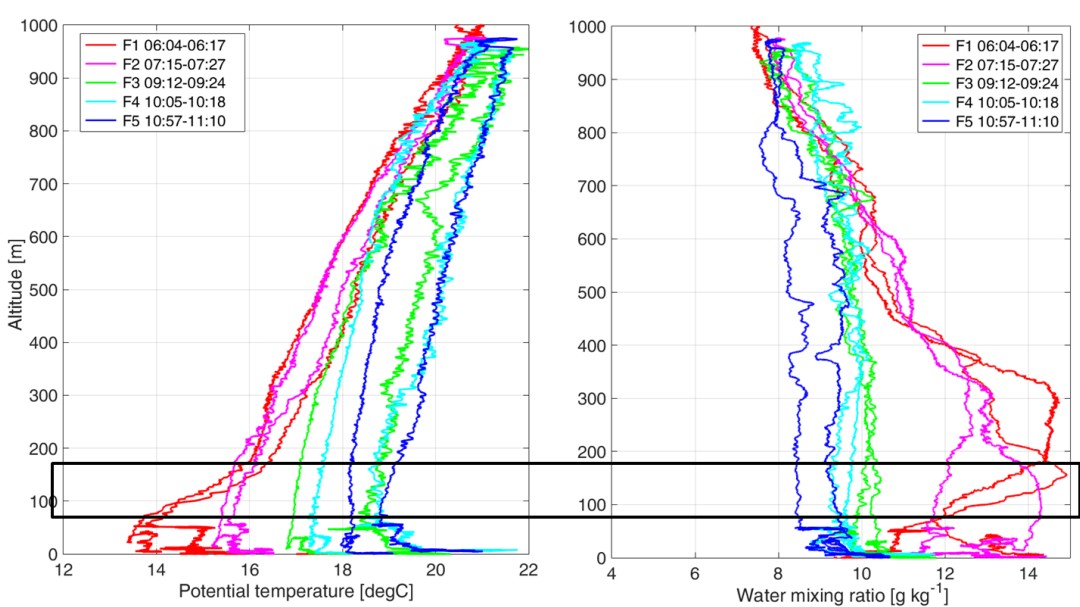

**Figure 8.** Profiles of potential temperature and water vapour mixing ratio obtained on 5 September 2018. The times of the three flights are given in UTC. The horizontal box represents the height interval of the temperature inversion, which is also visible in the large changes of the water vapour mixing ratio.

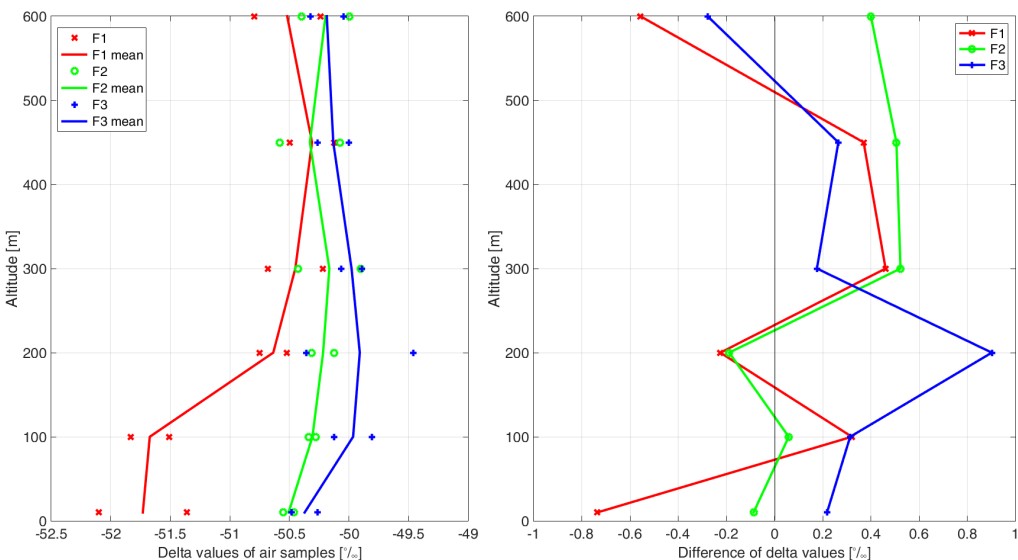

**Figure 9.** Profiles of a) $\delta^{13}C$ values for all air samples during the flights on 23 May 2018, and b) difference of delta values between the double samples filled simultaneously at the same altitude.

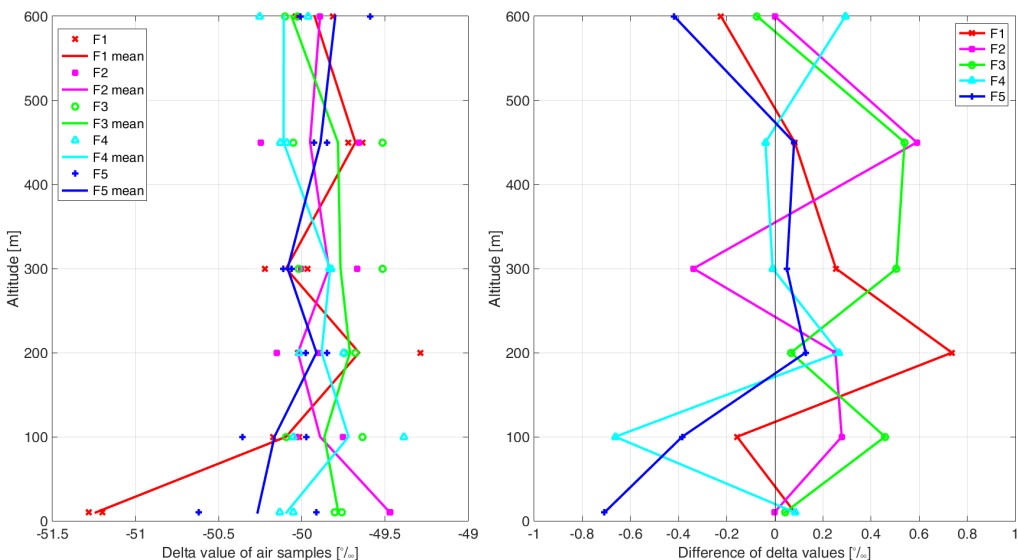

**Figure 10.** Profiles of a) $\delta^{13}C$ values for all air samples during the flights on 5 September 2018, and b) difference of delta values between the double samples filled simultaneously at the same altitude.

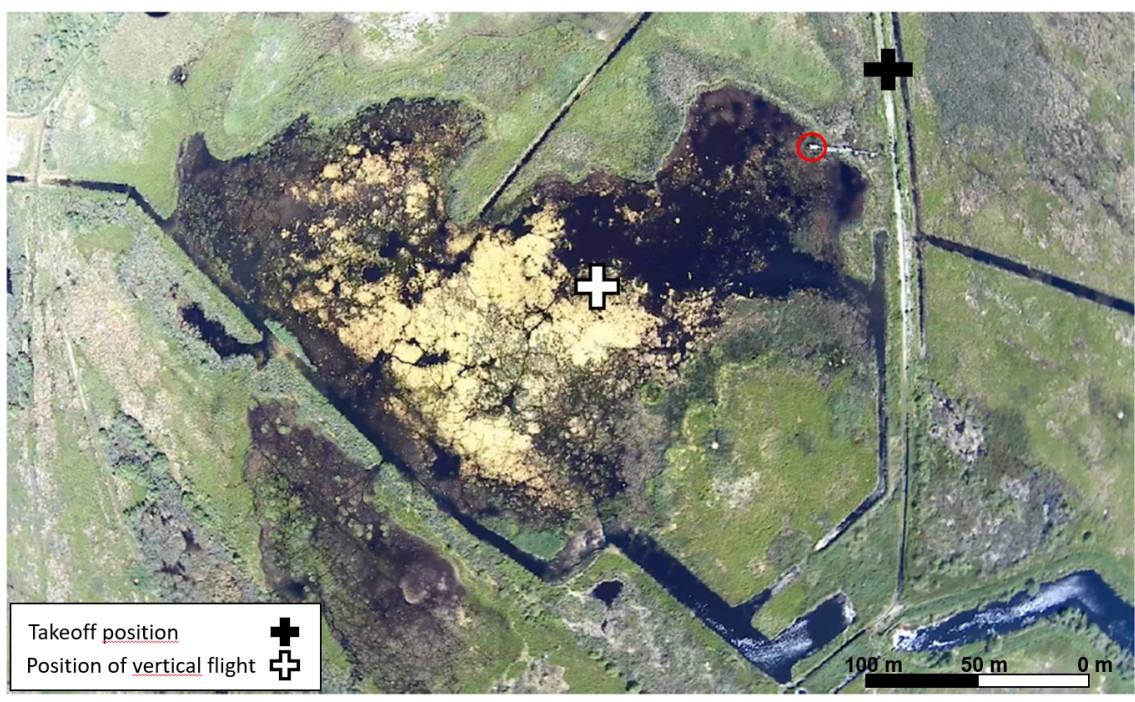

**Figure 11.** Aerial picture of the polder Zarnekow obtained with the quadrocopter ALICE on 23 May 2018. The location of the EC station is indicated with a red circle.

**Table 1.** Water samples on 5 September 2018: location and water depth of sampling, colour, concentration and delta value.

| Number | location | water depth | colour | conc. [ppm] | delta value [‰] |
|---|---|---|---|---|---|
| Z-1 | Peene River influx, surface water, 2 m from shore | 5-10 cm | middle yellow | 0.339 | -30.5 |
| Z-2 | Peene River influx, deep water, 2 m from shore | 40-50 cm | middle yellow | 0.246 | -30.1 |
| Z-3 | Polder Zarnekow, surface water, pond, 30 cm from edge | less than 5 cm | strong yellow | 2312.7 | -42.8 |
| Z-4 | Polder Zarnekow, surface water, pond | 5-10 cm | strong yellow | 4765.2 | -48.2 |
| Z-5 | Trench in the East, 1 m from shore, between plants | 5-10 cm | light yellow | 111.38 | -5.1 |
| Z-6 | Trench in the East, near shore | 5-10 cm | light yellow | 2397.9 | -49.2 |