# Peer review of "Studying boundary layer methane isotopy and vertical mixing processes at a rewetted peatland site by unmanned aircraft system"

_Atmospheric Measurement Techniques, 2019_

## Referee Comment (RC1) · Anonymous Referee #1 · 17 Jul 2019

The manuscript presents a quadrocopter equipped with a flask sampler in a proof of concept study looking at delta 13C in methane above a peatland on two mornings. Using unmanned aircraft for sampling in the lower atmosphere is not new. Comparable approaches are referred to in the introduction. The novelty here is perhaps the target parameter: delta 13C in methane.

Major issues

Precision of the isotope measurement in the laboratory is indicated to be about 0.5 permill (page 5, line 28). Could this indication be more precise? Are the 0.5 permill one sigma for the same, repeatedly measured sample? Considering the difference in

isotopic signal between nocturnal boundary layer and the air above during the first flight of a day is only about 1 permill, the isotopic signal of the polder is barely significant. Its size may be limited by three factors. First, the difference may be small in isotopic signature between the polder and other, presumably also biogenic sources in the larger surroundings. Second, the polder area may be too small to substantially alter atmospheric methane isotopic composition in the lower tens of metres, especially at wind speeds of several metres per second. A simple box model would suffice for an initial estimate. Third, the flights were performed well after sunrise. Much of the methane trapped near the surface during the night had already been mixed to greater altitude, as can be seen in the methane concentration in Figure 9. The data in Figures 9 and 10 seems to include erroneous measurements (e.g. zero methane around 18:00 on 5 September and a large number of spikes going above 0.3 mmol/m3 on both days).

Page 9, last paragraph: "The hypothesis ..." is not plausible at all. Horizontal gradients in air decrease with altitude because of higher windspeeds and more efficient mixing with increasing height above ground. Any remaining gradient will certainly not be large enough to cause measurable differences in samples taken simultaneously "only 13 cm apart". The sampling container is filled "within less than 2 s" (page 4, line 32). A horizontal wind speed of 2 m/s already results in a sample integrating air over a few metres in the horizontal direction. In addition, the turbulence caused by the rotors will add several metres across which the sample integrates in the vertical dimension. Hence, the "difference" between samples taken in parallel is pure noise caused by differences in the tightness of the sample containers, in handling, and in the analysis in the laboratory. I wonder how this issue can have escaped a group of nine authors.

Minor issues

Abstract Line 10: isotopic signature of what? delta13C or D/H ?

Page 2, line 6: "Yet, current knowledge of CH4 sources remains inadequate." What do you mean by "inadequate"? Inadequate to justify or guide mitigation measures? I do

not think so. We know very well where methane is produced and what could be done to reduce emissions.

What is the size of the sample containers (flasks)? Also, more detail about the valves and their connection with the glass flasks is necessary.

I do not understand the sentence on page 4, line 33-34 ("The most critical point were the manual plastic valves deployed routinely for the glass flasks in open position.")

What is the vertical resolution of sampling, given the turbulence caused by four rotors keeping 19 kg of vehicle and payload afloat?

Page 6, line 11: "The restoration of the peatland area towards a net sink of greenhouse gases, and in particular CH4, ..." Why should a rewetted peatland turn into a sink of methane?

What is the size of the rewetted Polder Zarnekow and other peatlands in its vicinity?

The number of Figures is too large for a short article.

---

## Referee Comment (RC2) · Anonymous Referee #2 · 22 Jul 2019

The study by Lampert et al. describes an approach by which air samples can be obtained from different altitudes reaching far into the atmospheric boundary layer (ABL) using an unmanned aerial system. Though the higher level ideas and potential benefits behind the determination of CH4 isotopic composition within the ABL are distributed throughout the manuscript, a concise summary of the higher level aims of this study in both abstract and introduction is missing. In addition, there is inadequate use of terms with regard to isotopic compositions and isotope ratios, as well as imprecision in describing footprint sizes. After considering the below specified aspects, this publication could be reconsidered for publication in AMT.

[Figure]

1. At this stage, the abstract is not convincing and stops short of revealing the scientific benefit that may arise from profiles of methane isotopic composition reaching far into the atmospheric boundary layer. There is a more or less recent publication by Röckmann et al. in ACP that summarizes nicely the potential benefit of tall tower and, thus, probably also airborne measurements. I suggest using this overarching view to introduce the topic and made a suggestion in this direction in the abstract- section of this referee comment. Some of the ideas can also be found in the conclusions section.
2. In the methods section, I am missing information on from where exactly air is sampled. There is quite some discussion on the adverse effect air parcel transport due to rotor downwash, so that this issue, and how it was tackled (if at all), should be mentioned already in the methods section. 3. In the methods section, water samples are referred to as a proxy for spatial heterogeneity of the emitted methane isotopic composition. However, methane dissolved in water and the methane emitted to the atmosphere will have a different isotopic composition due to the fractionation effect of volatilization. I wonder why the authors have not taken air samples from a closed chamber several times and used the keeling-plot approach to determine the isotopic composition of the emitted methane? This would be a direct measurement of the isotopic composition of emitted methane.
4. Possible influence from surrounding sources, i.e. surrounding land cover/land use, is not discussed at all, but is necessary in view of the highest altitudes at which measurements took place.
5. When on page 4, the design point is mentioned, and the reasoning behind, the introduction makes more sense. Please underline this connection already in the introduction.
6. There is a fundamental misunderstanding of isotope ratios. The isotope ratio (not isotopy ratios) cannot be negative. After transformation of the isotope ratios to the delta scale, negative values arise if the abundance of the heavy atom is lower than that of the reference material. The use of technical terms in a scientific journal has to be right.

7. There seems to be quite some imprecision when the discussion addresses footprints. I suggest consulting the literature, one example is given in the detailed comments. It is unclear why footprints for a measurement taken at 100 m should be similarly small as for a measurement at 10 m, as long as stable atmospheric conditions prevail.

8. Especially the photographs in figure 2 are at least inadequate for a journal like AMT. Both the picture detail and the background makes it practically impossible to discern anything. In addition, the font color is illegible.

9. Figure 3 is not helpful for the manuscript.

10. Figure 9: showing concentration until 10:00 would be enough, source isotopic composition may be best reflected during night. What is the increased concentration at approx. 12:00?

See some more detailed comments below.

Title

ok

Abstract

P1L1: A higher level rationale is missing in the abstract. In addition, the first sentence sounds a bit like quadrocopter air sampling influences laboratory analysis. I get the drift of the authors, but suggest something like "The determination of the methane isotopic composition at a tall tower and in high temporal resolution indicated a high potential to further constrain methane budgets at regional scales. However, tall towers are rare research infrastructures that may be supported by airborne measurement approaches. In this proof of concept study, we demonstrate the feasibility of using a quadrocopter to obtain air samples at heights between 10 and 600 m above ground. The methane isotopic composition of the air samples was subsequently determined in the laboratory. ..." Introduction

P2L1: I suggest changing to "... 32 times that of CO2 ..."

P2L6: Please elaborate on what exactly remains inadequate. The source categories

are quite clear I think, but the relative contributions of the different categories need better constraints.

P2L20: please revise typo in background

P2L21: I suggest changing to "indicates", since isotopic compositions of biological sources and fossil / thermogenic methane may overlap in the region of -55 per mil.

Figure 2: please use a monochrome background for the figure, and change to "battery" on left picture

P3L17: please decide either for methane isotopic composition or for isotope ratio. Isotopic ratio is at least uncommon. In addition, line 16 starts naming the aim, and in line 17, the goals are listed. Please revise the section.

P3L22: I suggest sharing your hypotheses that this is due to the more pronounced influence of the wetland producing depleted CH4 compared to the atmospheric background in absence of turbulent mixing. See comment above with regard to isotopic ratio ←→ isotope ratio

Materials and Methods

P4L2: it sounds like the quadrocopter was not based on a commercial chassis. If this applies, please state more clearly.

P4L3: unclear why the payload should alter the dimensions of the UAV. Please provide details. P4L4: I suggest: "At a tare weight of 4 kg, ALICE's maximum take-off weight is 25kg." Does the tare weight include batteries for driving the rotors, or are the LiPo batteries for both operating the scientific instrumentation and driving the rotors?

P4L27-31: I don't understand exactly, if there are both manual and magnetic valves necessary. Please explain in more detail. My first impression would be that only magnetic valves that are normally closed would suffice the purpose?

P4L34: From this sentence I guess that I am right that there are two sets of valves. However, the meaning of this sentence is unclear.

P5L4-6: this section seems redundant to me. The electromagnetic valves were explained some lines above.

P5L16: I suggest starting with where the samples were analysed and that they were brought there directly after the end of the mission, and then go into detail. Otherwise the transport issue pops up out of the blue.

P6L11: I cannot follow that a restored peatland should become a net sink for methane?

P6L25: I suggest including subsection heads "site description", "water sampling" and "flight strategy" P6L35: Please provide rationale for sampling during descent only.

P7L5: Please explain why the first flights were after sunrise though the aim was to investigate the transition from night time stable conditions to daytime turbulence.

Results

P7L11/12: please change typo to evidenced, suggest to change to "indicated"

P7L22: Please indicate nocturnal temperature inversion in figure.

Discussion

P8L25: please change to stratification

P9L9: -48to -49 is not an isotope ratio, but the delta value. I suggest using the term isotopic composition throughout the manuscript.

P9L26: This assumption is erroneous. The footprint of a measurement at a given height is larger under stable conditions, because the effect of advective transport is much more distinct compared to a situation dominated by turbulence in which vertical mixing is stronger. Please refer to several papers for example by Kljun et al., e.g., "A three-dimensional backward lagrangian footprint model for a wide range of boundary layer stratifications". In addition, the footprints of measurements at 10m height will be much smaller than those at 100m height.

---

## Author Comment (AC1) · 18 Sep 2019

**Answers to referees on* "Studying boundary layer methane isotopy and vertical mixing processes at a rewetted peatland site by unmanned aircraft system"**

Astrid Lampert[1], Falk Pätzold[1], Magnus O. Asmussen[1], Lennart Lobitz[1], Thomas Krüger[1], Thomas Rausch[1], Torsten Sachs[1,2], Christian Wille[2], Denis Sotomayor Zakharov[3], Dominik Gaus[3], Stephan Bansmer[3], and Ellen Damm[4]

[1]TU Braunschweig, Institute of Flight Guidance, Hermann-Blenk-Str. 27, 38108 Braunschweig, Germany
[2]German Research Centre for Geosciences, Telegrafenberg, 14473 Potsdam, Germany
[3]TU Braunschweig, Institute of Fluid Mechanics, Hermann-Blenk-Str. 37, 38108 Braunschweig, Germany
[4]Alfred Wegener Institute, Helmholtz Centre for Polar and Marine Research, Am Handelshafen 12, 27570 Bremerhaven, Germany

**Correspondence:** Astrid Lampert (Astrid.Lampert@tu-braunschweig.de)

**1   Answers to referees**

The authors would like to thank the anonymous referee for the comments on the manuscript. In the following, the comments are given in *italic*. The answers are given in normal letters. The modified text in the manuscript is given in quotation marks.

**2   Referee 1**

5 *The manuscript presents a quadrocopter equipped with a flask sampler in a proof of concept study looking at delta 13C in methane above a peatland on two mornings. Using unmanned aircraft for sampling in the lower atmosphere is not new. Comparable approaches are referred to in the introduction. The novelty here is perhaps the target parameter: delta 13C in methane.*

We changed the beginning of the abstract to "The combination of two well-established methods, of quadrocopter-borne air
10 sampling and of methane isotopic analyses, is applied to determine the origin of methane at different altitudes and to study mixing processes. A proof of concept study was performed to demonstrate the capabilities of quadrocopter air sampling for subsequently analysing the methane isotopic composition $\delta^{13}$C in the laboratory. The advantage of the system compared to classical sampling at ground and at tall towers is the flexibility concerning sampling location, and in particular the flexible choice of sampling altitude, allowing to study layering and mixing of air masses with potentially different origin of methane."

15 *Precision of the isotope measurement in the laboratory is indicated to be about 0.5 permill (page 5, line 28). Could this indication be more precise? Are the 0.5 permill one sigma for the same, repeatedly measured sample?*

We changed the text to: "Precision, determined as the maximum difference of delta values during repeated analysis of ambient Bremerhaven air samples taken simultaneously and analysed consecutively many times per year, is better than 0.5‰. "

*Considering the difference in isotopic signal between nocturnal boundary layer and the air above during the first flight of a day is only about 1 permill, the isotopic signal of the polder is barely significant.*

We agree that the isotopic signature of the air surrounding the polder is similar. This can be expected as the rewetted area is larger than the polder. Further, the methane isotopic signature from ruminants is similar, as summarised by Röckmann et al. (2016). We added in the abstract: "The systematically more negative delta values occurred only as long as the nocturnal temperature inversion was present."

*Its size may be limited by three factors. First, the difference may be small in isotopic signature between the polder and other, presumably also biogenic sources in the larger surroundings.*

We agree with this point, and changed the text in the discussion to "The difference in delta values obtained for air samples near ground and above the temperature inversion during stable stratification is around 1.5‰, thus significantly higher than the uncertainties (flight 1 for each measurement day). This shows that the observed systematic differences are real measurement features, not measurement uncertainty. Other methane sources in the surroundings of the polder are presumably of biological origin as well, they may include larger areas of the rewetted peatland and ruminants, with similar isotopic composition (Röckmann et al., 2016)."

*Second, the polder area may be too small to substantially alter atmospheric methane isotopic composition in the lower tens of metres, especially at wind speeds of several metres per second.*

During the night, when the temperature inversion inhibits mixing and the locally emitted methane is trapped, the wind speed was below $2\,\mathrm{m\,s^{-1}}$. Therefore, the effect of a different isotopic composition below the temperature inversion can be traced back to local emissions. We changed the discussion text to: "Under stable atmospheric conditions and low wind speed smaller than $2\,\mathrm{m\,s^{-1}}$ during the night and in the early morning hours, the near-surface methane concentration is enhanced, as no vertical mixing is possible, in agreement with Emeis (2008); Brosy et al. (2017)."

*A simple box model would suffice for an initial estimate.*

We do not agree that a box model is helpful for this case. As underlined by the very different methane concentrations in the water, we assume local inhomogeneities of sources. On the contrary, a suitable method for qunatifying small and inhomogeneous sources is urgently needed. We changed the text in the conclusion section to: "The differences in delta values of water and air, the differences in delta values between both flight days and the development during each day emphasize the highly complex and inhomogeneous nature of methane processes on horizontal scales below 1 km in sediments, at the sediment-water and the water-atmosphere interface. Therefore, a suitable method is required for quantifying small-scale inhomogeneous methane sources."

*Third, the flights were performed well after sunrise. Much of the methane trapped near the surface during the night had already been mixed to greater altitude, as can be seen in the methane concentration in Figure 9.*

We agree with this point, the methane concentration was higher during the night, as indicated in Fig. 9. We changed the text in the discussion to: "Under stable atmospheric conditions and low wind speed smaller than $2 \, \text{m s}^{-1}$ during the night and in the early morning hours, the near-surface methane concentration is enhanced, as no vertical mixing is possible, in agreement with Emeis (2008); Brosy et al. (2017). This can be seen in the time series of the methane concentrations for both days as well

5   (Figs. 9 and 10). During the flights, the methane concentration was already smaller again, which can be explained by vertical mixing up to the temperature inversion of around $100 \, \text{m}$. This dilusion also influences the isotopic composition."

*The data in Figures 9 and 10 seems to include erroneous measurements (e.g. zero methane around 18:00 on 5 September and a large number of spikes going above 0.3 mmol/m3 on both days).*

Originally, we used only the data set of the LICOR 7700. Now we corrected the data with the quality controlled closed path
10   data set of the Los Gatos sensor. We further included the time series in the figure of the diurnal cycle of the meteorological parameters.

*Page 9, last paragraph: "The hypothesis ..." is not plausible at all. Horizontal gradients in air decrease with altitude because of higher windspeeds and more efficient mixing with increasing height above ground. Any remaining gradient will certainly not be large enough to cause measurable differences in samples taken simultaneously "only 13 cm apart". The sampling container*
15   *is filled "within less than 2 s" (page 4, line 32). A horizontal wind speed of 2 m/s already results in a sample integrating air over a few metres in the horizontal direction.*

We added in the text information about the sample integration:

"Assuming that in the worst case the sampling takes place within the downwash of the rotor blades of not more than $-19 \, \text{m s}^{-1}$, the sampling time of $1.3 \, \text{s}$ duration results in a vertical resolution of around $25 \, \text{m}$. Sampling during descent with a speed of
20   $-2.5 \, \text{m s}^{-1}$ adds an uncertainty in the altitude of $5 \, \text{m}$. Altogether, the sampling is influenced by air in a height interval of $30 \, \text{m}$. This is the sufficient for the sampling intervals of around $100 \, \text{m}$, and for determining that the sampling was done below or above the temperature inversion."

Further, we added in the discussion: "In the presented case, the parallel samples obtained at $10 \, \text{m}$ altitude seem to be mostly of the same origin with small differences in the delta value. Above, differences in the delta values are higher. A wind speed of
25   less than $2 \, \text{m s}^{-1}$ during the sampling results in horizontal sample integration over not more than $4 \, \text{m}$. Small-scale differences in methane isotopic composition can be introduced by mixing processes, and may be reinforced by mixing induced by the quadrocopter system. A high variability of methane sources is in agreement with the highly variable methane water concentrations measured within a radius of $100 \, \text{m}$ on 5 September 2018." A visualisation of such small-scale mixing processes is shown in Figs. 5 and 6. In this case, colour was released artificially near ground (inhomogeneous sources). The homogeneous layer
30   building up due to stable stratification was disturbed by turbulence, in this case induced by the aircraft, and large horizontal gradients are visible as a result.

*In addition, the turbulence caused by the rotors will add several metres across which the sample integrates in the vertical dimension.*

We included three more co-authors who have done numerical simulations, and added a section about simulations of the flow
35   induced by a propeller blade:

"In order to quantify the effect of the vertical flow induced by the quadrocpter, numerical simulations were performed with the software ANSYS CFX. The simulations were transient in nature using a Reynolds-Averaged Navier Stokes (RANS) approach with the Shear Stress Transport (SST) turbulence model (Menter, 1994). A simplified model of the propeller blade was used, with a multidomain approach: The blade is enclosed in a rotating domain, surrounded by a static domain. Simulations were performed for hover with a propeller rotation speed of $3167\,\mathrm{min}^{-1}$, for vertical climb at a speed of $6.5\,\mathrm{m\,s}^{-1}$ with a rotation speed of $3913\,^{-1}$ and of vertical descent at a speed of $-2.5\,\mathrm{m\,s}^{-1}$ with a rotation speed of $2880\,\mathrm{min}^{-1}$. An ambient temperature of $0°\,\mathrm{C}$ and pressure of $1023\,\mathrm{hPa}$ were considered. Contours of relative vertical velocity show a core region of positive relative velocity directly below the center of the blade, and a negative relative velocity up to $19\,\mathrm{m\,s}^{-1}$ below the blade for a distance exceeding $0.75\,\mathrm{m}$ (Fig. 2). Additionally, zones of recirculation can be seen around the tips of the propeller, especially for the descent case. The air sampling system is contained in the middle of the copter, and is less affected by artificial turbulence than the areas below the rotor blades.

Assuming that in the worst case the sampling takes place within the downwash of the rotor blades of not more than $-19\,\mathrm{m\,s}^{-1}$, the sampling time of $1.3\,\mathrm{s}$ duration results in a vertical resolution of around $25\,\mathrm{m}$. Sampling during descent with a speed of $-2.5\,\mathrm{m\,s}^{-1}$ adds an uncertainty in the altitude of $5\,\mathrm{m}$. Altogether, the sampling is influenced by air in a height interval of $30\,\mathrm{m}$. This is the sufficient for the sampling intervals of around $100\,\mathrm{m}$, and for determining that the sampling was done below or above the temperature inversion."

Further, we added in the discussion section: "In the presented case, the parallel samples obtained at $10\,\mathrm{m}$ altitude seem to be mostly of the same origin with small differences in the delta value. Above, differences in the delta values are higher. A wind speed of less than $2\,\mathrm{m\,s}^{-1}$ during the sampling results in horizontal sample integration over not more than $4\,\mathrm{m}$ at the sampling altitude. Small-scale differences in methane isotopic composition can be introduced by mixing processes, and may be reinforced by mixing induced by the quadrocopter system. A high variability of methane sources is in agreement with the highly variable methane water concentrations measured within a radius of $100\,\mathrm{m}$ on 5 September 2018."

*Hence, the "difference" between samples taken in parallel is pure noise caused by differences in the tightness of the sample containers, in handling, and in the analysis in the laboratory. I wonder how this issue can have escaped a group of nine authors.*

We do not agree with the referee. The tightness of each of the sample containers was controlled by a pressure sensor, and the pressure data were both provided in real-time to the operators and recorded as well. Sample containers were air tight until being filled with air. The smaller pressure due to sampling above ground level was checked before closing the manual valves. Further, it cannot be assumed that the methane concentration was homogeneous. As an illustration, Fig. 5 and 6 show artificially released colour in a stably stratified morning atmosphere. Large horizontal concentration gradients can exist, and it is particularly difficult to quantify such non-stationary cases, as described by Schaller et al. (2018).

Finally, the difference between the delta values of the samples is height dependent, with much lower differences at the lowest sampling altitude, and an apparent temporal development of the difference for the sampling at higher altitudes. Pure noise should follow a more stochastic distribution of the differences in delta values.

*Abstract Line 10: isotopic signature of what? delta13C or D/H ?*

We changed the text to "methane isotopic composition $\delta^{13}\mathrm{C}$"

*Page 2, line 6: "Yet, current knowledge of CH4 sources remains inadequate." What do you mean by "inadequate"? Inadequate to justify or guide mitigation measures? I do not think so. We know very well where methane is produced and what could be done to reduce emissions.*

We totally agree with the referee. The text is now changed to "Yet, current knowledge of $CH_4$ biogeochemical processes, transport and small-scale distribution remains inadequate."

*What is the size of the sample containers (flasks)? Also, more detail about the valves and their connection with the glass flasks is necessary.*

We added a figure explaining details of the valves and included in the text: "The air sampling system consists of 12 glass flasks (sample containers) of 100 ml content"

*I do not understand the sentence on page 4, line 33-34 ("The most critical point were the manual plastic valves deployed routinely for the glass flasks in open position.")*

We explain now more in detail and added Fig. 1. The text has been changed to: "The air sampling system consists of 12 glass flasks (sample containers) of 100 ml content, which are evacuated before take-off. They are equipped with two manual valves, one on each side, and additionally one electromagnetic valve, which is applied only during the flight (Fig. 1). Directly before the mission, each glass flask is linked with a vacuum pump RE5 of Vacuubrand, Germany. One valve is left open, and an electromagnetic valve is connected, which is normally closed. Then the flask is evacuated, and the pressure is controlled by a pressure sensor integrated in the electromagnetic valve. The flasks are opened during the flight with magnet valves, that are triggered either manually by remote control or automatically at altitudes predefined by the operator. After triggering, ambient pressure is reached within less than 2 s. The pressure sensors integrated in the valves are used to monitor air tightness. The most delicate component are the manual plastic valves, used to close the glass flasks for transport, which are designed to be air tight when closed, but not when open. They had to be treated individually and controlled to make sure that no leakage occurred during the mission. For quality control and redundance, two glass flasks were filled simultaneously, resulting in six possible sampling altitudes during one flight."

*What is the vertical resolution of sampling, given the turbulence caused by four rotors keeping 19 kg of vehicle and payload afloat?*

We added a figure of simulations showing the vertical velocity induced by the propeller blades (Fig. 2) and added in the text: "Assuming that in the worst case the sampling takes place within the downwash of the rotor blades of not more than -19 m s$^{-1}$, the sampling time of not more than 2 results in a vertical resolution of around 40 m. Sampling during descent with a speed of -2.5 m s$^{-1}$ adds an uncertainty in the altitude of 5 m. Altogether, the sampling is influenced by air in a height interval of 45 m. This is the sufficient for the sampling intervals of around 100 m, and for determining that the sampling was done below or above the temperature inversion."

*Page 6, line 11: "The restoration of the peatland area towards a net sink of greenhouse gases, and in particular CH4, ..." Why should a rewetted peatland turn into a sink of methane?*

35   This is indeed wrong. The idea was to point out that the restoration of a peatland area takes some time to act as $CO_2$ sink. Further, the restoration is usually accompanied by enhanced methane emissions, and it takes some years until the system is not such a strong methane source any more. We changed the text to:

"The restoration of the peatland area towards a net sink of the greenhouse gas $CO_2$ is a process of several years to decades. Initially, the restoration is accompanied by a strong increase in $CH_4$ emissions, which depend on vegetation and the water level

5   (Couwenberg et al., 2011; Zak et al., 2015)."

*What is the size of the rewetted Polder Zarnekow and other peatlands in its vicinity?*

We included in the text: " The total rewetted area is 421 ha in size (**?**)."

*The number of Figures is too large for a short article.*

We removed the original Fig. 3 and Fig. 14. We further suggest to combine Fig. 4 and Fig. 13 in the final publication.

**10 References**

Brosy, C., Krampf, K., Zeeman, M., Wolf, B., Junkermann, W., Schäfer, K., Emeis, S., and Kunstmann, H.: Simultaneous multicopter-based air sampling and sensing of meteorological variables, Atmos. Meas. Tech., 10, 2773-2784, 2017.

Couwenberg, J., Thiele, A., Tanneberger, F., Augustin, J., Bärisch, S., Dubovik, D., Liashchynskaya, N., Michaelis, D., Minke, M., Skura-tovich, A., and Joosten, H.: Assessing greenhouse gas emissions from peatlands using vegetation as a proxy, Hydrobiologia, 674, 67-89, 2011.

Emeis, S.: Examples for the determination of turbulent (sub-synoptic) fluxes with inverse methods, Meteorol. Z., 17, 1, 003-011, 2008.

Gelbrecht, J., Zak, D., and Augustin, J. (eds.): Phosphor- und Kohlenstoff-Dynamik und Vegetationsentwicklung in wiedervernässten Mooren des Peenetals in Mecklenburg-Vorpommern, Status, Steuergrößen und Handlungsmöglichkeiten, report 26/2008 of Leibniz-Institute of Freshwater Ecology and Inland Fisheries, 101 pp., 2008.

Menter, F.R.: Two-equation eddy-viscosity turbulence models for engineering applications, AIAA Journal, 32, 8, 1598-1605, 1994.

Röckmann, T., Eyer, S., van der Veen, C., Popa, M.E., Tuzson, B., Monteil, G., Houweling, S., Harris, E., Brunner, D., Fischer, H., Zazzeri, G., Lowry, D., Nisbet, E.G., Brand, W.A., Necki, J.M., Emmenegger, L., and Mohn, J.: In situ observations of the isotopic composition of methane at the Cabauw tall tower site, Atmos. Chem. Phys., 16, 10469-10487, 2016.

Schaller, C., Kittler, F., Foken, F., and Gödecke, M.: Characterisation of short-term extreme methane fluxes related to non-turbulent mixing above an Arctic permafrost ecosystem, Atmos. Chem. Phys. Discuss., https://doi.org/10.5194/acp-2018-277, 2018.

Zak, D., Reuter, H., Augustin, J., Shatwell, T., Barth, M., Gelbrecht, J., and McInnes, R.J.: Changes of the $CO_2$ and $CH_4$ production potential of rewetted fens in the perspective of temporal vegetation shifts, Biogeosciences, 12, 2455-2468, 2015.

[Figure]

**Figure 1.** ALICE vital components.

[Figure]

**Figure 2.** Simulation of the flow induced by a propeller blade during hover, climb and descent.

[Figure]

**Figure 3.** Diurnal course of the main meteorological parameters air temperature, global radiation, wind speed, wind direction, and methane concentration recorded at the meteorological mast at Zarnekow on 23 May 2018.

[Figure]

**Figure 4.** Diurnal course of the main meteorological parameters air temperature, global radiation, wind speed, wind direction, and methane concentration recorded at the meteorological mast at Zarnekow on 05 September 2018.

[Figure]

**Figure 5.** Illustration of concentration during stable stratification. The photo was taken at the air field Aue/Hattorf, Germany, on 2 October 2011 at 16:15 UTC, copyright Institute of Flight Guidance, TU Braunschweig

[Figure]

**Figure 6.** Illustration of small-scale concentration variability induced by aircraft. The photo was taken at the air field Aue/Hattorf, Germany, on 2 October 2011 at 16:15 UTC, copyright Institute of Flight Guidance, TU Braunschweig

---

## Author Comment (AC2) · 18 Sep 2019

**Answers to referees on* "Studying boundary layer methane isotopy and vertical mixing processes at a rewetted peatland site by unmanned aircraft system"**

Astrid Lampert[1], Falk Pätzold[1], Magnus O. Asmussen[1], Lennart Lobitz[1], Thomas Krüger[1], Thomas Rausch[1], Torsten Sachs[1,2], Christian Wille[2], Denis Sotomayor Zakharov[3], Dominik Gaus[3], Stephan Bansmer[3], and Ellen Damm[4]

[1]TU Braunschweig, Institute of Flight Guidance, Hermann-Blenk-Str. 27, 38108 Braunschweig, Germany
[2]German Research Centre for Geosciences, Telegrafenberg, 14473 Potsdam, Germany
[3]TU Braunschweig, Institute of Fluid Mechanics, Hermann-Blenk-Str. 37, 38108 Braunschweig, Germany
[4]Alfred Wegener Institute, Helmholtz Centre for Polar and Marine Research, Am Handelshafen 12, 27570 Bremerhaven, Germany

**Correspondence:** Astrid Lampert (Astrid.Lampert@tu-braunschweig.de)

**1   Answers to referees**

The authors would like to thank the anonymous referee for the comments on the manuscript. In the following, the comments are given in *italic*. The answers are given in normal letters. The modified text in the manuscript is given in quotation marks.

**2   Referee 2**

5  *The study by Lampert et al. describes an approach by which air samples can be obtained from different altitudes reaching far into the atmospheric boundary layer (ABL) using an unmanned aerial system. Though the higher level ideas and potential benefits behind the determination of CH4 isotopic composition within the ABL are distributed throughout the manuscript, a concise summary of the higher level aims of this study in both abstract and introduction is missing.*

We changed the beginning of the abstract to

10  "The combination of two well-established methods, of quadrocopter-borne air sampling and of methane isotopic analyses, is applied to determine the origin of methane at different altitudes and to study mixing processes. A proof of concept study was performed to demonstrate the capabilities of quadrocopter air sampling for subsequently analysing the methane isotopic composition $\delta^{13}$C in the laboratory. The advantage of the system compared to classical sampling at ground and at tall towers is the flexibility concerning sampling location, and in particular the flexible choice of sampling altitude, allowing to study

15  layering and mixing of air masses with potentially different origin of methane."

We added in the introduction: "The need to improve understanding of the heterogeneous methane source and the transition from the surface into the atmosphere in the Arctic motivated the development of a flexible airborne sampling system, which provides information on atmospheric stability."

*In addition, there is inadequate use of terms with regard to isotopic compositions and isotope ratios, as well as imprecision in describing footprint sizes.*

We checked the text for consistency in wording. Now the terms "isotopic composition" and "delta values" are used throughout the text.

Further, we now avoid the term "footprint", which is used as a specific technical term. We changed the interpretation section to:

"The isotopic composition of the two air samples taken on 5 September 2018 simultaneously but with a constant horizontal distance of 13 cm agree within 0.1‰ at the lowest altitude of 10 m for Flight 1 to 4 (Fig. **??**). Besides this strong locally and temporally related agreement of the isotopic composition, for other altitudes and flights this difference is larger than the uncertainty, and therefore the systematic differences are treated as features. There are several striking features in the profiles of isotopic composition, from lower to higher altitudes:

- On 5 September 2018 the difference in delta values between the two simultaneous samples is systematically smaller at 10 m altitude compared to the higher altitudes, except the last profile.

- On 5 September 2018 the differences in delta values at 100 m altitude increase during the course of the day. The profiles of differences in delta values exhibit similarities for parts of the profile between subsequent flights on both days.

- The delta values are more negative in the morning before vertical mixing starts, as long as a temperature inversion is present (first flight on 23 May 2018 below 150 m, and first flight on 5 September 2018 below 70 m). This is in agreement with methane from biologic processes emitted from the surface that are not vertically mixed.

As the order of analysing the air samples was chosen randomly, the differences of the delta values exceed the uncertainty and the differences in delta values correlate in section between subsequent flights, it is assumed that the differences in delta values are physically present in the air samples. Two aspects can be highlighted:

- An ideally vertically stratified delta value would not be sampled by the present system as the very dynamic circulation process around the copter does not result in a homogenously mixed air at the sample ports. On the contrary, this circulation process can even amplify natural inhomogeneity. It is assumed that the differences in delta values indicate natural inhomogeneity, but it is not possible to prove it based on the data set.

- Beside the buoyancy vertical turbulent mixing the natural inhomogeneity of delta values is not known for the measurement site. Small-scale horizontal variability can be induced by inhomogeneous sources. Episodic $CH_4$ outbursts on short time scales of few min have been observed by Schaller et al. (2018). The high spatial and temporal variability of methane concentration and isotopic composition reported here is in agreement with their observations. Such variability

of methane emissions at the field site as well as the potential upwind $CH_4$ sources cause the inhomogeneous character of the air samples.

Respecting that the air sample profiling gives a snapshot of a turbulent mixing process, a clear transition in the vertical distribution of the delta values can be seen."

5    *After considering the below specified aspects, this publication could be reconsidered for publication in AMT.*

We hope that we can answer all concerns satisfactorily.

*1. At this stage, the abstract is not convincing and stops short of revealing the scientific benefit that may arise from profiles of methane isotopic composition reaching far into the atmospheric boundary layer.*

We changed the beginning of the abstract to: "To determine the source of methane emissions at different altitudes and to study
10    mixing processes, a quadrocopter air sampling system was developed. A proof of concept study was performed to demonstrate the capabilities of quadrocopter air sampling for analysing the methane isotopic composition $\delta^{13}C$ in the laboratory. The advantage of the system compared to classical sampling at ground is the flexibility concerning sampling location, and in particular the flexible choice of sampling altitude, allowing to study layering and mixing of air masses with potentially different origin of methane."

15    *There is a more or less recent publication by Roeckmann et al. in ACP that summarizes nicely the potential benefit of tall tower and, thus, probably also airborne measurements.*

We would like to thank the referee for pointing out this interesting publication. We included the following sentence in the introduction: "Field measurements of methane isotopic composition have been performed at the Cabauw tower at a sampling altitude of 20 m, demonstrating the potential of isotopic analyses to determine contributions from isotopically different sources
20    to the dominating source (Röckmann et al., 2016)." Further, we added "UAS can be operated in remote areas, requiring less infrastructure in comparison with permanent measurement stations, and they can be used more flexibly than manned aircraft, enabling fast reactions to environmental events like changes of emissions through rain, drought, construction, or fire."

*I suggest using this overarching view to introduce the topic and made a suggestion in this direction in the abstract- section of this referee comment. Some of the ideas can also be found in the conclusions section.*

25    We would like to thank the referee for the practical recommendation, which we took into account (see answer to comment above).

*2. In the methods section, I am missing information on from where exactly air is sampled. There is quite some discussion on the adverse effect air parcel transport due to rotor downwash, so that this issue, and how it was tackled (if at all), should be mentioned already in the methods section.*

30    We included three more co-authors who have done numerical simulations, and added a subsection about simulations with the following text: "In order to quantify the effect of the vertical flow induced by the quadrocpter, numerical simulations were performed with the software ANSYS CFX. The simulations were transient in nature using a Reynolds-Averaged Navier Stokes (RANS) approach with the SST turbulence model (Menter, 1994). A simplified model of the propeller blade was used, with a multidomain approach: The blade is enclosed in a rotating domain, surrounded by a static domain. Simulations were per-
35    formed for hover with a propeller rotation speed of $3167\,\mathrm{min}^{-1}$, for vertical climb at a speed of $6.5\,\mathrm{m\,s}^{-1}$ with a rotation speed

of 3913 $^{-1}$ and of vertical descent at a speed of -2.5 m s$^{-1}$ with a rotation speed of 2880 min$^{-1}$. An ambient temperature of 0° C and pressure of 1023 hPa was considered. Contours of relative vertical velocity show a core region of positive relative velocity directly below the center of the blade, and a negative relative velocity up to 19 m s$^{-1}$ below the blade for a distance exceeding 0.75 m (Fig. 2). Additionally, zones of recirculation can be seen around the tips of the propeller, especially for the

5    descent case. The air sampling system is contained in the middle of the copter, and is less affected by artificial turbulence than the areas below the rotor blades.

Assuming that in the worst case the sampling takes place within the downwash of the rotor blades of not more than -19 m s$^{-1}$, the sampling time of 1.3 s duration results in a vertical resolution of around 25 m. Sampling during descent with a speed of -2.5 m s$^{-1}$ adds an uncertainty in the altitude of 5 m. Altogether, the sampling is influenced by air in a height interval of 30 m.

10    This is the sufficient for the sampling intervals of around 100 m, and for determining that the sampling was done below or above the temperature inversion."

*3. In the methods section, water samples are referred to as a proxy for spatial heterogeneity of the emitted methane isotopic composition. However, methane dissolved in water and the methane emitted to the atmosphere will have a different isotopic*

15    *composition due to the fractionation effect of volatilization. I wonder why the authors have not taken air samples from a closed chamber several times and used the keeling-plot approach to determine the isotopic composition of the emitted methane? This would be a direct measurement of the isotopic composition of emitted methane.*

We agree that it would be good to have closed chamber samples for comparison of the delta value. However, the aim of this study is to show the vertical distribution of the methane isotopic composition in dependence of atmospheric stability, which we

20    interpret as related to different methane sources.

*4. Possible influence from surrounding sources, i.e. surrounding land cover/land use, is not discussed at all, but is necessary in view of the highest altitudes at which measurements took place.*

We agree that we cannot fully understand the processes without taking into account more information, like land cover. However, the aim of the manuscript is to show that it is possible to identify vertical layers of different isotopic composition

25    with the experimental setup of the quadrocopter. Further, the case study shows that there is a high horizontal variability of methane sources, which requires specific tools to further understand the observations. We changed the introduction to: "The goal of the study is two-fold:

- Proof-of-concept for the experimental setup of the quadrocopter borne sampling system, and subsequent laboratory analyses, to identify vertical layers of different isotopic composition

30    - Identification of small-scale atmospheric methane inhomogeneities which require the development of new methods for understanding dynamic processes

In order to test the system's capabilities of providing reliable vertical profiles of the isotopic composition, measurements were performed at a rewetted peatland site, Polder Zarnekow (Zerbe et al., 2013), which is known as a source of biologically produced methane **?**. In the absence of turbulent mixing, local emissions of the wetlands produce a depleted delta value

compared to the atmospheric background above the temperatue inversion. During the morning transition, when the stable stratification is gradually replaced by a convectively mixed atmospheric boundary layer, the isotopic composition should adjust to a constant delta value throughout the profile within the uncertainties of laboratory air sample isotopic analyses. To support the hypothesis of the small-scale horizontal inhomogeneity at the study site, the methane concentration of water samples from locations within a radius of 100 m was analysed."

*5. When on page 4, the design point is mentioned, and the reasoning behind, the introduction makes more sense. Please underline this connection already in the introduction.*

We added earlier in the introduction: "The need to improve understanding of the heterogeneous methane source and the transition from the surface into the atmosphere in the Arctic motivated the development of a flexible airborne sampling system, which provides information on atmospheric stability."

*6. There is a fundamental misunderstanding of isotope ratios. The isotope ratio (not isotopy ratios) cannot be negative. After transformation of the isotope ratios to the delta scale, negative values arise if the abundance of the heavy atom is lower than that of the reference material. The use of technical terms in a scientific journal has to be right.*

We apologize for the sloppy use of the terms. Now the terms "isotopic composition" and "delta values" are used in the text accordingly.

*7. There seems to be quite some imprecision when the discussion addresses footprints. I suggest consulting the literature, one example is given in the detailed comments. It is unclear why footprints for a measurement taken at 100 m should be similarly small as for a measurement at 10 m, as long as stable atmospheric conditions prevail.*

We admit that we used the technical term "footprint" in an imprecise way, and rephrased the sentences without using it.

*8. Especially the photographs in figure 2 are at least inadequate for a journal like AMT. Both the picture detail and the background makes it practically impossible to discern anything. In addition, the font color is illegible.*

We replaced the picture with Fig. 1.

*9. Figure 3 is not helpful for the manuscript.*

We removed the figure and the reference in the text.

*10. Figure 9: showing concentration until 10:00 would be enough, source isotopic composition may be best reflected during night. What is the increased concentration at approx. 12:00?*

We corrected the plots and included them in the figures of the time series of meteorological parameters.

*P1L1: A higher level rationale is missing in the abstract.*

We changed the beginning of the abstract to: "To determine the source of methane emissions at different altitudes and to study mixing processes, a quadrocopter air sampling system was developed. A proof of concept study was performed to demonstrate the capabilities of quadrocopter air sampling for subsequently analysing the methane isotopic composition $\delta^{13}$C in the laboratory. The advantage of the system compared to classical sampling at ground is the flexibility concerning sampling location, and in particular the flexible choice of sampling altitude, allowing to study layering and mixing of air masses with potentially different origin of methane."

*In addition, the first sentence sounds a bit like quadrocopter air sampling influences laboratory analysis. I get the drift of the authors, but suggest something like "The determination of the methane isotopic composition at a tall tower and in high temporal resolution indicated a high potential to further constrain methane budgets at regional scales. However, tall towers are rare research infrastructures that may be supported by airborne measurement approaches. In this proof of concept study,*

5 *we demonstrate the feasibility of using aquadrocopter to obtain air samples at heights between 10 and 600 m above ground. The methane isotopic composition of the air samples was subsequently determined in the laboratory."*

We added in the introduction: "The need to improve understanding of the heterogeneous methane source and the transition from the surface into the atmosphere in the Arctic motivated the development of a flexible airborne sampling system, which provides information on atmospheric stability. In this context, unmanned aerial systems (UAS) fill an observational gap for

10 methane mixing processes. They are able to sample small scales with a typical horizontal distance of 1 km, if they are required to be operated in the line of sight, and they reach the top of the atmospheric boundary layer, with a maximum altitude of typically around 1 km. UAS can be operated in remote areas, requiring less infrastructure in comparison with permanent measurement stations, and they can be used more flexibly than manned aircraft, enabling fast reactions to environmental events like changes of emissions through rain, drought, construction, or fire."

*P2L1: I suggest changing to "32 times that of CO2"*

We changed as suggested.

*P2L6: Please elaborate on what exactly remains inadequate. The source categories are quite clear I think, but the relative contributions of the different categories need better constraints.*

20 We totally agree with the referee. The text is now changed to "Yet, current knowledge of $CH_4$ biogeochemical processes remains inadequate."

*P2L20: please revise typo in background*

We corrected the typo.

*P2L21: I suggest changing to "indicates", since isotopic compositions of biological sources and fossil / thermogenic*

25 *methane may overlap in the region of -55 per mil.*

We changed the text as suggested.

*Figure 2: please use a monochrome background for the figure, and change to "battery" on left picture*

We replaced the figure and changed as suggested (see Fig. 1)

*P3L17: please decide either for methane isotopic composition or for isotope ratio. Isotopic ratio is at least uncommon.*

30 We now use the terms "isotopic composition" and "delta value" consistently throughout the manuscript.

*In addition, line 16 starts naming the aim, and in line 17, the goals are listed. Please revise the section.*

We revised the section, and changed to:

"The goal of the study is two-fold:

- Proof-of-concept for the experimental setup of the quadrocopter borne sampling system, and subsequent laboratory
35 analyses, to identify vertical layers of different isotopic composition

– Identification of small-scale atmospheric methane inhomogeneities which require the development of new methods for understanding dynamic processes

"

*P3L22: I suggest sharing your hypotheses that this is due to the more pronounced influence of the wetland producing depleted*
5   *CH4 compared to the atmospheric background in absence of turbulent mixing.*

We changed the text to: "In the absence of turbulent mixing, local emissions of the wetlands produce a depleted delta value compared to the atmospheric background above the temperature inversion."

*See comment above with regard to isotopic ratio - isotope ratio*
10   We now use the term "isotopic composition" throughout the text.

*P4L2: it sounds like the quadrocopter was not based on a commercial chassis. If this applies, please state more clearly.*

We changed the text to "The quadrocopter ALICE was designed as platform to carry meteorological sensors and 12 glass bottles for air sampling. The construction of the quadrocopter was calculated for the specific tasks and payload described in the following. Therefore, all relevant load cases that were expected during the flight were applied in analytical and numerical
15   models to optimize the structure of the quadrocopter. Modern manufacturing methods like selective laser sintering and laser cutting where used to build the structure as light-weight as possible but as stable as necessary."

*P4L3: unclear why the payload should alter the dimensions of the UAV. Please provide details.*

We changed the sentence to "ALICE has dimensions of 1.56 m x 1.56 m x 0.38 m, not including the scientific payload."

*P4L4: I suggest: "At a tare weight of 4 kg, ALICE's maximum take-off weight is 25kg." Does the tare weight include batteries*
20   *for driving the rotors, or are the LiPo batteries for both operating the scientific instrumentation and driving the rotors?*

We changed the text to "At a tare weight of 4 kg, ALICE's maximum take-off weight is 25 kg. For the operations presented here, the total weight was 19 kg, which is composed of 4 kg the quadrocopter system itself, 7.2 kg of LiPo batteries with a total capacity of 21 Ah and a nominal voltage of 44.4 V for rotor power supply, and 7.8 kg payload including sensors, glass bottles, data acquisition, power supply for payload and a safety parachute of $12\,\mathrm{m}^2$."

25   *P4L27-31: I don't understand exactly, if there are both manual and magnetic valves necessary. Please explain in more detail. My first impression would be that only magnetic valves that are normally closed would suffice the purpose?*

We replaced the figure of the ALICE system and included more information about the valves (see Fig. 1).

*P4L34: From this sentence I guess that I am right that there are two sets of valves. However, the meaning of this sentence is unclear.*
30   We hope that the setting of valves becomes clear with the new figure (Fig. 1).

*P5L4-6: this section seems redundant to me. The electromagnetic valves were explained some lines above.*

We agree that this is already mentioned before, and removed the sentence.

*P5L16: I suggest starting with where the samples were analysed and that they were brought there directly after the end of the mission, and then go into detail. Otherwise the transport issue pops up out of the blue.*

We changed the order of the sentences to "Following the quadrocopter mission, the sample containers (SC) were transported to the laboratory at the Alfred Wegener Institute in Bremerhaven, Germany for aanalysis of the isotopic composition. The $\delta^{13}C$ value of the air samples was analyzed using a Delta plus XP mass spectrometer combined with a combustion oven, a gas pressure interface and a pre-concentration device (PreCon) (ThermoFinnigan, Bremen, Germany)."

5     *P6L11: I cannot follow that a restored peatland should become a net sink for methane?*

This is indeed wrong. The intention was to point out that the restoration of a peatland area takes some time to act as $CO_2$ sink. Further, the restoration is usually accompanied by enhanced methane emissions, and it takes some years until the system is not such a strong methane source any more. We changed the text to:

"The restoration of the peatland area towards a net sink of the greenhouse gas $CO_2$ is a process of several years to decades.
10   Initially, the restoration is accompanied by a strong increase in $CH_4$ emissions, which depend on vegetation and the water level (Couwenberg et al., 2011; Zak et al., 2015)."

*P6L25: I suggest including subsection heads "site description", "water sampling" and "flight strategy"*

We modified the structure as suggested.

*P6L35: Please provide rationale for sampling during descent only.*

15   We added in the text: "During ascent, the temperature profiles were studied, as a base to plan sampling altitudes for the descent."

*P7L5: Please explain why the first flights were after sunrise though the aim was to investigate the transition from night time stable conditions to daytime turbulence.*

We added in the text: "Flights were permitted between sunrise and sunset. As it takes some hours until the nocturnal tem-
20   perature inversion is heated away, it was possible to take the air samples during the transition from nocturnal stable boundary layer to the convectively mixed boundary layer. "

*P7L11/12: please change typo to evidenced, suggest to change to "indicated"*

We changed to "indicated" as suggested.

*P7L22: Please indicate nocturnal temperature inversion in figure.*

25   We indicate the height interval of the temperature inversion with a horizontal box in the figures of potential temperature and water vapour mixing ratio.

*P8L25: please change to stratification*

We changed the text as suggested.

*P9L9: -48to -49 is not an isotope ratio, but the delta value. I suggest using the term isotopic composition throughout the*
30   *manuscript.*

We now use the terms "isotopic composition" and "delta value" consistently throughout the text.

*P9L26: This assumption is erroneous. The footprint of a measurement at a given height is larger under stable conditions, because the effect of advective transport is much more distinct compared to a situation dominated by turbulence in which vertical mixing is stronger. Please refer to several papers for example by Kljun et al., e.g., "A three-dimensional backward*

*lagrangian footprint model for a wide range of boundary layer stratifications". In addition, the footprints of measurements at 10 m height will be much smaller than those at 100 m height.*

We admit that we used the technical term "footprint" in an imprecise and misleading way, and rephrased the sentences without using it.

**References**

Brosy, C., Krampf, K., Zeeman, M., Wolf, B., Junkermann, W., Schäfer, K., Emeis, S., and Kunstmann, H.: Simultaneous multicopter-based air sampling and sensing of meteorological variables, Atmos. Meas. Tech., 10, 2773-2784, 2017.

Couwenberg, J., Thiele, A., Tanneberger, F., Augustin, J., Bärisch, S., Dubovik, D., Liashchynskaya, N., Michaelis, D., Minke, M., Skuratovich, A., and Joosten, H.: Assessing greenhouse gas emissions from peatlands using vegetation as a proxy, Hydrobiologia, 674, 67-89, 2011.

Emeis, S.: Examples for the determination of turbulent (sub-synoptic) fluxes with inverse methods, Meteorol. Z., 17, 1, 003-011, 2008.

Gelbrecht, J., Zak, D., and Augustin, J. (eds.): Phosphor- und Kohlenstoff-Dynamik und Vegetationsentwicklung in wiedervernässten Mooren des Peenetals in Mecklenburg-Vorpommern, Status, Steuergrößen und Handlungsmöglichkeiten, report 26/2008 of Leibniz-Institute of Freshwater Ecology and Inland Fisheries, 101 pp., 2008.

Menter, F.R.: Two-equation eddy-viscosity turbulence models for engineering applications, AIAA Journal, 32, 8, 1598-1605, 1994.

Röckmann, T., Eyer, S., van der Veen, C., Popa, M.E., Tuzson, B., Monteil, G., Houweling, S., Harris, E., Brunner, D., Fischer, H., Zazzeri, G., Lowry, D., Nisbet, E.G., Brand, W.A., Necki, J.M., Emmenegger, L., and Mohn, J.: In situ observations of the isotopic composition of methane at the Cabauw tall tower site, Atmos. Chem. Phys., 16, 10469-10487, 2016.

Sachs, T., Koebsch, F., Franz, D., Larmanou, E., Serafimovich, A., Kohnert, K., Jurasinski, G., and Augustin, J.: Mehr Moor? Zur treibhausgasdynamik wiedervernässter Feuchtgebiete, System Erde, 5, 1, DOI: 10.2312/GFZ.syserde.05.01.4, 6pp., 2015.

Schaller, C., Kittler, F., Foken, F., and Gödecke, M.: Characterisation of short-term extreme methane fluxes related to non-turbulent mixing above an Arctic permafrost ecosystem, Atmos. Chem. Phys. Discuss., https://doi.org/10.5194/acp-2018-277, 2018.

Zak, D., Reuter, H., Augustin, J., Shatwell, T., Barth, M., Gelbrecht, J., and McInnes, R.J.: Changes of the $CO_2$ and $CH_4$ production potential of rewetted fens in the perspective of temporal vegetation shifts, Biogeosciences, 12, 2455-2468, 2015.

Zerbe, S., Steffenhagen, P., Parakenings, K., Timmermann, T., Frick, A., Gelbrecht, J., and Zak, D.: Ecosystem Service Restoration after 10 Years of Rewetting Peatlands in NE Germany, Environmental Management, 51, 1194-1209, 2013.

[Figure]

**Figure 1.** ALICE vital components.

[Figure]

**Figure 2.** Simulation of the flow induced by a propeller blade during hover, climb and descent.

[Figure]

**Figure 3.** Diurnal course of the main meteorological parameters air temperature, global radiation, wind speed, wind direction, and methane concentration recorded at the meteorological mast at Zarnekow on 23 May 2018.

[Figure]

**Figure 4.** Diurnal course of the main meteorological parameters air temperature, global radiation, wind speed, wind direction, and methane concentration recorded at the meteorological mast at Zarnekow on 05 September 2018.

[Figure]

**Figure 5.** Illustration of concentration during stable stratification. The photo was taken at the air field Aue/Hattorf, Germany, on 2 October 2011 at 16:15 UTC, copyright Institute of Flight Guidance, TU Braunschweig

[Figure]

**Figure 6.** Illustration of small-scale concentration variability induced by aircraft. The photo was taken at the air field Aue/Hattorf, Germany, on 2 October 2011 at 16:15 UTC, copyright Institute of Flight Guidance, TU Braunschweig

---

## Author Response (AR2)

**Answers to referees on* "Studying boundary layer methane isotopy and vertical mixing processes at a rewetted peatland site by unmanned aircraft system"**

Astrid Lampert[1], Falk Pätzold[1], Magnus O. Asmussen[1], Lennart Lobitz[1], Thomas Krüger[1], Thomas Rausch[1], Torsten Sachs[1,2], Christian Wille[2], Denis Sotomayor Zakharov[3], Dominik Gaus[3], Stephan Bansmer[3], and Ellen Damm[3]

[1]TU Braunschweig, Institute of Flight Guidance, Hermann-Blenk-Str. 27, 38108 Braunschweig, Germany
[2]German Research Centre for Geosciences, Telegrafenberg, 14473 Potsdam, Germany
[3]TU Braunschweig, Institute of Fluid Mechanics, Hermann-Blenk-Str. 37, 38108 Braunschweig, Germany
[4]Alfred Wegener Institute, Helmholtz Centre for Polar and Marine Research, Am Handelshafen 12, 27570 Bremerhaven, Germany

**Correspondence:** Astrid Lampert (Astrid.Lampert@tu-braunschweig.de)

**1  Answers to referees**

The authors would like to thank the two anonymous referees and the editor for their comments on the manuscript. In the following, the comments are given in *italic*. The answers are given in normal letters. The modified text in the manuscript is given in quotation marks.

5  ## 2  Editor

*Associate Editor Decision: Reconsider after major revisions (10 Nov 2019) by Christof Ammann Comments to the Author:*
*Dear authors,*
*the two Referees have reviewed the revised version of your manuscript and found significant improvements. However, according to their assessment (see the individual referee reports in the manuscript overview) there are still important shortcomings*
10  *that need further improvement. In addition, I have identified some issues as listed in the following comments.*
*1. I consider the micrometeorological interpretation as insufficient and partly problematic. The objective of the study (as stated in the introduction) was to identify the isotopic composition of vertical layers in the boundary layer and to test the capability of the system to provide reliable vertical profiles of the isotopic composition. The authors should consider that a vertical profile in the micrometeorological sense is always a time-averaged profile over typical averaging intervals of about 30 min.*

We are not convinced that the classical time-averaging over 30 min is suitable to capture turbulent small-scale processes for spatially heterogeneous sources and non-stationary signals, as changes may occur on much smaller temporal and spatial

scales. E.g. Schaller et al. (2018) showed that methane is frequently emitted from oversaturated groundwater or surfacewater in eposidic outbursts of few minutes duration, which is triggered by atmospheric events. Such events cannot be treated by the commonly applied 30 min averaging.

We added in the introduction:

"However, the classical time-averaging over 30 min is not suitable to capture turbulent small-scale processes for spatially heterogeneous sources and non-stationary signals, as changes may occur on much smaller temporal and spatial scales. Alternatives are e.g. simultaneous measurements upwind and downwind of heterogeneous methane emitting sources **?**. Schaller et al. (2018) showed that methane is frequently emitted from oversaturated ground water or surface water in eposidic outbursts of few minutes duration, which is triggered by atmospheric events. Such events cannot be treated by the commonly applied 30 min averaging."

*In this way, the turbulent fluctuations (temporally and spatially) can be integrated. Alternatively to a temporal average, a spatial average (on constant heights) may yield the same result, but only if the surface is sufficiently homogeneous.*

*Therefore, the authors should consider and discuss, whether the instantaneous point sampling is sufficient and useful for micrometeorological profile analysis. It should be discussed whether a longer sampling time (e.g. using a capillary at the inlet, e.g. with repeated valve openings) would be more useful and possible with the sampling platform.*

We added in the discussion section:

"For a systematic comparison with the eddy flux measurements, temporally and spatially integrated measurements would be adequate. However, averaging is only suitable for sufficiently homogeneous surface conditions and emissions. This is not the case here, as already indicated by the different methane concentrations and isotopic compositions for the water samples. Therefore, the instantaneous point samples cannot be compared directly with the classical micrometeorological methods like 30 min averaged EC analyses. For further investigating small-scale inhomogeneities, new methods for observations and analyses are required: Instantaneous profiles of the methane concentration and isotopic composition upwind and downwind of an EC tower could be combined with wavelet analyses instead of EC covariance analyses, as suggested by Schaller et al. (2018)."

*2. The issue addressed in comment 1 also has implications for the interpretation of the differences between the two parallel samples. I do not think that this instantaneous and small-scaled differences represent "systematic features". If they are not due to sampling/analysis problems, they represent local turbulent fluctuations. But two instantaneous samples are probably not enough to provide insight into the turbulence dynamics. Please consider this in the formulation of the second objective in Section 1 and in the discussion in Section 4.*

We do agree with the editor that the differences in two parallel samples are very likely caused by local turbulent fluctuations. We changed the text in the introduction to:

"Besides this strong locally and temporally related agreement of the isotopic composition, for other altitudes and flights this difference is in the range of the uncertainty. A possible explanation for the systematic differences is a combination of spatial

inhomogeneities and turbulent mixing processes. There are several striking patterns in the profiles of isotopic composition, from lower to higher altitudes:

- On 5 September 2018 the difference in delta values between the two simultaneous samples is systematically smaller at 10 m altitude (much better than the uncertainty) compared to the higher altitudes, except the last profile.

- On 5 September 2018 the differences in delta values at 100 m altitude increase during the course of the day. The profiles of differences in delta values exhibit similarities for parts of the profile between subsequent flights on both days.

- The delta values are significantly more negative in the morning before vertical mixing starts, as long as a temperature inversion is present (first flight on 23 May 2018 below 150 m, and first flight on 5 September 2018 below 70 m). The difference between delta values below and above the temperature inversion is larger than the uncertainty. This is in agreement with methane from biologic processes emitted from the surface that are not vertically mixed."

*3. page 8, line 15/16: Indicate the (horizontal) position of the UAV profile measurements in Fig. 4 and/or Fig. 11*

We did as suggested (see Fig. 4 and Fig. 3).

*4. The newly introduced Fig. 3 is quite interesting but I have the following suggestions for improvement: a) I am not sure if the indicated sequence in the figure caption "hover, climb, descent" corresponds to the A, B, C sequence of the figure panels. Please give precise attribution in the Figure caption.*

We attributed the three sub-figures to the description in the caption.

*b. It would be useful to indicate the position of the sample inlets relative to the propeller blades and the simulated induced flow fields.*

We included in the text more information about the distance between inlets and propeller blades: "The air sampling system consists of 12 glass flasks (sample containers) of 100 ml volume, which are evacuated before take-off. Their arrangement with respect to the copter can be seen in Fig. 1. The distance from the tip of the closest rotor blades to the inlet is approximately 7 cm."

*c. Indicate the (simulated) ascent and descent speed in the Figure caption.*

We changed the figue caption to: "Simulation of the flow induced by a propeller blade during hover (A), vertical climb at a speed of $6.5\,\mathrm{m\,s^{-1}}$ (B) and descent at a speed of $-2.5\,\mathrm{m\,s^{-1}}$ (C)."

*5. According to Figs. 5 and 6, the wind direction during all flights was between 0 and 90 deg corresponding to north-easterly wind directions. This would mean that the Eddy Covariance tower measured flux footprints outside of the polder area shown in Figs. 4 and 11 (and distant from the water sampling). Please comment on that.*

We included in the text: "As the eddy flux tower is located at the North-East of the polder, flux footprints are influenced by areas outside of the polder area (see Figs. 4 and 3). As the profile measurements were performed directly above the polder, the sampled air has a different footprint. The locations of the water samples are all outside the footprint area of the EC tower."

*6. Add length scales to Figs. 4 and 11. This is important to relate the horizontal extension of the polder to the profile measurement heights.*

We did as suggested (see Fig. 4 and Fig. 3).

*Please consider the above comments and the referee comments (see separate reports) in your further revision.*

The answers to the referees' comments are provided in the following.

**3  Referee 1**

*The authors have done a good job in revising the manuscript. Most problems of the earlier version have been solved. The only major issue remaining is that differences in delta 13C between samples taken simultaneously at a horizontal distance of 0.13 m on a descending quadrocopter a few 100 m above ground level are considered "features" (page 11, lines 6-9).*

We see that the difference in the delta values close to the surface is much smaller than the difference in the delta values at higher altitudes of few 100 m. The differences are in the same orer of magnitude as the uncertainty caused by the measurement and analysis process. Therefore, we conclude that the larger differences at higher altitudes can be explained by higher turbulent mixing compared to the samples obtained close to the surface. We rephrased the paragraph, avoiding the use of the term "features":

"Besides this strong locally and temporally related agreement of the isotopic composition, for other altitudes and flights this difference is in the range of the uncertainty. A possible explanation for the systematic differences is a combination of spatial inhomogeneities and turbulent mixing processes. There are several striking patterns in the profiles of isotopic composition, from lower to higher altitudes:"

*Air movement above the inversion layer is not laminar but turbulent.*

We totally agree with the referee. In our opinion, this is the main difference between the sampling altitudes of 10 m and above. Above, we have higher differences in isotopic composition, which we think is caused by higher turbulent mixing.

*Further, turbulence next to the parallel sample inlets is enhanced by the four propellers surrounding the sampler.*

Again we do agree, and we even did a first quantification with numerical simulations.

*Horizontal sample integration due to natural wind is up to 4 m (page 10, lines 24 and 25). Vertical integration during descent at 2 m/s is at least 2.6 m (2m/s * 1.3 s). Even without accounting for the additional flow and mixing induced by the propellers, only a minute fraction of the sample volumes collected in parallel can originate from different air parcels that have not been subject to mixing since they had been in contact with different methane sources at the land surface more than 100 m below the copter.*

We are not convinced that this perfect mixing can be assumed for strong vertical or horizontal gradients. As an illustration, we include again the illustrations of coloured air during stable stratification, which is then mixed by an aircraft (Figs. 5 and 6). Here, strong local gradients of colour concentration are clearly visible.

*Let's be optimistic and say that 10% of the volumes sampled in parallel were from different, unmixed air parcels. Then, the difference in delta 13C between these parcels would have had to be 4 permil to produce a 0.4 permil difference between the parallel samples. I find such horizontal gradients more than 100 m above ground highly implausible. The difference between below and above inversion layer is less than half that value. Although the source of methane on the ground may vary by a few permil across distances of a few metres, the probability that such a difference is conserved in air during transport across and well beyond the inversion layer is certainly much smaller than is the probability of obtaining by pure chance the results shown in 10b. Anything smaller than 0.5 permil is anyway within the limits of uncertainty (precision level) and therefore not a "feature". To have a chance of eventually observing a real horizontal gradient, samples would need to be taken upwind and downwind of a strong local source (e.g. the rewetted peatland). An example of horizontal concentration gradients between up- and downwind a local CH4 source is shown in Denmead et al (Atmos. Environ. 32, 3679–3688, 1998).*

This is indeed an interesting calculation. We modified Sect. 4.2, emphasizing more the uncertainty of the analyses of 0.5‰: "The isotopic composition of the two air samples taken on 5 September 2018 simultaneously but with a constant horizontal distance of 13 cm agree within 0.1‰ at the lowest altitude of 10 m for Flight 1 to 4 (Fig. **??**). This is a much better agreement than the uncertainty of 0.5‰. However, this value has been determined experimentally for long laboratory time series, and may be much better for subsequent analyses. Besides this strong locally and temporally related agreement of the isotopic composition, for other altitudes and flights this difference is in the range of the uncertainty. A possible explanation for the systematic differences is a combination of spatial inhomogeneities and turbulent mixing processes. There are several striking patterns in the profiles of isotopic composition, from lower to higher altitudes:

- On 5 September 2018 the difference in delta values between the two simultaneous samples is systematically smaller at 10 m altitude (much better than the uncertainty) compared to the higher altitudes, except the last profile.

- On 5 September 2018 the differences in delta values at 100 m altitude increase during the course of the day. The profiles of differences in delta values exhibit similarities for parts of the profile between subsequent flights on both days.

- The delta values are significantly more negative in the morning before vertical mixing starts, as long as a temperature inversion is present (first flight on 23 May 2018 below 150 m, and first flight on 5 September 2018 below 70 m). The difference between delta values below and above the temperature inversion is larger than the uncertainty. This is in agreement with methane from biologic processes emitted from the surface that are not vertically mixed.

Although the differences of the delta values only slightly exceed the uncertainty, there are indications that the differences in delta values are physically present in the air samples: The order of analysing the air samples was chosen randomly and the differences in delta values show a similar height profile for subsequent flights."

Further, we mention the publication of **?** in the introduction:

"However, the classical time-averaging over 30 min is not suitable to capture turbulent small-scale processes for spatially heterogeneous sources and non-stationary signals, as changes may occur on much smaller temporal and spatial scales. Alternatives are e.g. simultaneous measurements upwind and downwind of heterogeneous methane emitting sources (**?**)."

*The rather peculiar argument continues on the same page, lines 21 to 24:*

"*An ideally vertically stratified delta value would not be sampled by the present system as the very dynamic circulation process around the quadrocopter does not result in a homogenously mixed air at the sample ports on the quadrocopter. On the contrary, this circulation process can even amplify natural inhomogeneity. It is assumed that the differences in delta values indicate natural inhomogeneity...*"

*How should natural inhomogeneities be amplified by circulation processes, such as visualised in Fig. 3? Is entropy in a fluid decreased by stirring? Can I un-mix coffee and milk by stirring it?*

I think there is a misunderstanding, as the referee and the authors have different perceptions of the mixing process in mind. In the perspective of the authors, we are thinking of the beginning of mixing. So yes, if we have coffee with milk and start stirring, initially we obtain turbulent structures with strong local gradients of brown and white structures. Probably the referee is more thinking of mixing as a finished process. Of course after some time, there is a homogeneous distribution of methane in an air parcel and of coffee with milk. But we think that we are watching the beginning of mixing processes here, where we do not have homogeneous mixing yet, as visualised in Figs. 5 and 6.

*Page 10, lines 18 and 20: "dilusion": do you mean "delusion" or "dilution"?*

Thank you for the comment. We changed to the more appropriate term "mixing"

*Please add a scale bar to Figures 4 and 11.*

A scale bar is now included in both figures.

**4    Referee 2**

5    *The study by Lampert et al. describes an approach by which air samples can be obtained from different altitudes reaching far into the atmospheric boundary layer (ABL) using an unmanned aerial system. The authors have added important information and I really appreciate the RANS modelling concerning rotor downwash. However, the authors have stopped short with fully exploiting the information provided, and turning them into valuable information for others in the discussion. For my understanding, this requires further elaboration. My most important points are as follows:*

10    *1. Still lack of precision in formulations, quite often I was confused at first and had to read several times to understand the meaning. For instance, the term origin sometimes refers to spatial allocation, and sometimes to the production mechanism. I suggest to use spatial origin and source process respectively.*

We exchanged the term "origin" with the more specific terms "spatial origin" and "source process" throughout the text.

*2. Determination of origin at different heights above ground is mentioned as one of the two most important aspects of the study this i) not the same aims as described in the introduction and ii) the abstract never gets back to a conclusion on this issue. The implications of aim 2 given in the introduction remain unclear. What new methods need to be developed?*

20    We modified the aims mentioned in the introduction slightly and adapt them to the aims mentioned in the abstract: "

– Proof-of-concept for the experimental setup of the quadrocopter borne sampling system, and subsequent laboratory analyses, to identify vertical layers of different isotopic composition and therefore the spatial origin and source processes

– Identification of small-scale atmospheric methane inhomogeneities which require the development of new methods for understanding turbulent mixing processes

25    "

Further, we changed the last part of the abstract to: "The airborne sampling system and consecutive analysis chain were shown to provide reliable and reproducible results for two samples obtained simultaneously. The method presents a powerful tool for distinguishing the source process of methane at different altitudes. The isotopic composition showed clearly depleted delta values directly above a biological methane source when vertical mixing was hampered by a temperature inversion, and different

30    delta values above, where the air masses originate from a different footprint area. The vertical distribution of methane isotopic composition can serve as tracer for mixing processes of methane within the atmospheric boundary layer."

Further, we included a new subsection "Improvement potential for multicopter based air sampling", where we discuss possible modification of the measurements and new methods for further analysing small-scale methane inhomogeneities.

*3. The RANS simulations are really interesting. Thank you for adding. However, it remains unclear to me how the calculation of the spatial allocation of the sampled air arises. Please clarify as I am not sure if I understood correctly: The authors have taken the largest relative air speed to calculate the travel distance during the 1.3 seconds it takes to fill the glass vessels. To my understanding, the sample air inlet was above the rotors, and I don't see these relative speeds above the rotors. This is*

5 *because air from a spherical half-space above the rotor is washed down by the rotor. The calculation would make sense if air exclusively straight above the rotors would be washed down, but the figure does not support this notion. It seems like the highest relative speed above the rotors is in the order of 12-15ms-1. Wouldn't it be more accurate to take the highest speed above the rotors to estimate the maximum travel distance?*

10 We added a more realistic calculation: "A more realistic estimate of the altitude interval uncertainty assumes the vertical velocities directly next to the rotor blade tips (See Fig. 1): The highest induced speed in close proximity of the tips of the rotor blades during descent is -8 m s$^{-1}$ (Fig. 2). This results in an altitude interval of around 10 m caused by the flow field plus an altitude interval of 5 m due to the vertical descent speed. Therefore, a realistic estimate of the altitude influencing the sampled air is 15 m."

*In addition it is not clear to me if only the downward pointing velocities are shown.*

We added in the figure caption: "Downward pointing velocities have a negative sign."

20 *Further, this figure suggests that having a sideways pointing tube as air inlet that reaches 25 cm beyond the rotor would be a good option for further uas based systems, or a tube that reaches 50 cm above the rotors.*

We included a separate subsection about improvement potential for multicopter air sampling:
"According to the simulations, undistorted air sampling with the multicopter is possible with a sideways pointing inlet that
25 reaches 25 cm beyond the rotor or a tube that reaches 50 cm above the rotors for hover and climb. Air sampling during descent experiences more additional disturbance by the rotor blades and therefore should be compared with air sampling during climb. The initial operation idea was to observe at first the atmospheric stratification in climb and determine the sampling altitudes for subsequent descent based on the altitude of the temperature inversion. However, the first simulation results quantify the difference in additional vertical velocities induced by the measurement system, which are much higher during descent (Fig. 2).
30 Due to efficiency reasons, the vertical climb speed is higher than the descent speed for the current ALICE system. The impact of the climb speed has to be taken into account for the temporal resolution of the sensors. In order to closer constrain the altitude interval of the sampled air, measurements during hover or slow climb flight in combination with an inlet tube of the dimensions mentioned above would be preferable for continuous sampling. However, for the presented sampling system with small volume, the air volume contained in the tubes is not exchanged continuously and would further induce uncertainties. Sampling during slow ascent or hover requires adjusting the battery capacities or the flight mission, e.g. the maximum flight

altitude. Further, simulations of the whole multicopter system including the payload are required to quantify the flow field and find the optimal sensor location.

5   For a systematic comparison with the eddy flux measurements, temporally and spatially integrated measurements would be adequate. However, averaging is only suitable for sufficiently homogeneous surface conditions and emissions. This is not the case here, as already indicated by the different methane concentrations and isotopic compositions for the water samples. Therefore, the instantaneous point samples cannot be compared directly with the classical micrometeorological methods like 30 min averaged EC analyses. For further investigating small-scale inhomogeneities, new methods for observations and analyses are

10   required: Instantaneous profiles of the methane concentration and isotopic composition upwind and downwind of an EC tower could be combined with wavelet analyses instead of EC covariance analyses, as suggested by Schaller et al. (2018)."

*Also the discussion if hovering would be the best option for taking a sample is not discussed at all. This is a pity and deserves a section in the discussion.*

This point is addressed in the new subsection (see comment above).

*4. Figure 1 doesn't tell anything*

20   We exchanged the figure with a different one showing the system on ground. Here, the location of the sampling system in relation to the platform is more clearly visible.

*See some more detailed comments below.*
*Title*
25 *ok*
*Abstract*
*P1L2: "quadrocopter-borne air sampling and of methane isotopic analyses, is applied to determine the origin of methane": I am not sure what the authors mean exactly with origin. Methane isotopic composition can give hints on the process that had produced the methane, but does not tell anything about the location. It could have been produced anywhere in the landscape*
30 *where the respective production process had taken place. In addition, I don't see where the authors get back to the origin of methane in the abstract. There should be a reference to the origin if this is mentioned at the foremost place in the abstract.*

xx

*Introduction P2L5: one reference can be removed*

We once removed the reference mentioned twice in the sentence.

*P2L19: Transformation of the isotope ratio to the delta scale results in values that allow to distinguish the source: Also the isotope ratios allow this process identification. The conversion to delta scale is just more intelligible for us, but is no other information. Just stick to : Isotopic composition allows to distinguish different source categories.*

True - we changed the sentence to "Isotopic composition allows to distinguish different source categories." as suggested.

*P2L24: please change from predominantly to "an increasing share of "*

We changed as suggested.

*P3L36: I suggest adding that this is typically happens during night.*

As suggested, we changed the sentence to: "In the absence of turbulent mixing, which typically happens during night, local emissions of the wetlands produce a depleted delta value compared to the atmospheric background above the temperature inversion."

*P5L15: please change from content to volume*

Ok.

*P8L8: heated away is colloquial, please rephrase*

We changed the sentence to: "As it takes some hours until the nocturnal temperature inversion is replaced by a well-mixed boundary layer with increasing solar radiation"

*Materials and Methods*
*Results*
*Table 1: the data shown is not isotope ratio but delta value.*

We corrected the term.

*Discussion P10L15: vertical mixing is possible, but hampered.*

We changed the sentence as suggested.

*L18: please change from dilusion to dilution. If the assumption is that air from above the inversion mixes with air below as the inversion climbs up and is dissolved, I suggest calling it mixing. Dilution alone, in the sence of mixing with for instance ch4 free air would not alter the isotopic composition. What you observed is the temporal variability of CH4 isotopic composition which is caused by transport and mixing.*

We agree that the term "mixing" is better suited.

*L20: dilusion again. Is this a real technical term that I am not aware of?*

15    We changed to mixing again.

*Conclusions: P12L5: in agreement with atmospheric stability: what does this mean? Maybe caused by atmospheric stability?*

We agree that this is confusing. We changed the sentence to: "With ALICE air samples and subsequent laboratory analyses, it is possible to determine differences in the methane isotopic composition caused by atmospheric stability."

*P12L16: isn't this bullet point the same as above?*

340

The referee is right, we removed this extra bullet point. We further changed the order of the bullet points, now commenting first on the measurement strategy, then on the double sampling, and finally, as an outlook, on the need of additional continuous onboard concentration measurements.

**345 References**

Schaller, C., Kittler, F., Foken, F., and Gödecke, M.: Characterisation of short-term extreme methane fluxes related to non-turbulent mixing above an Arctic permafrost ecosystem, Atmos. Chem. Phys. Discuss., https://doi.org/10.5194/acp-2018-277, 2018.

[Figure]

**Figure 1.** The quadrocopter ALICE before take-off in Zarnekow on 5 September 2018.

[Figure]

**Figure 2.** Simulation of the flow induced by a propeller blade during hover (A), vertical climb at a speed of 6.5 m s$^{-1}$ (B) and descent at a speed of -2.5 m s$^{-1}$ (C). Downward pointing velocities have a negative sign.

[Figure]

**Figure 3.** Aerial picture of the polder Zarnekow obtained with the quadrocopter ALICE on 5 September 2018. Almost the whole polder fell dry after the extremely warm and dry summer 2018. The sites where water samples were taken are indicated with Z1 to Z6. The location of the EC station is indicated with a red circle.

[Figure]

**Figure 4.** Aerial picture of the polder Zarnekow obtained with the quadrocopter ALICE on 23 May 2018. The location of the EC station is indicated with a red circle.

[Figure]

**Figure 5.** Illustration of concentration during stable stratification. The photo was taken at the air field Aue/Hattorf, Germany, on 2 October 2011 at 16:15 UTC, copyright Institute of Flight Guidance, TU Braunschweig

[Figure]

**Figure 6.** Illustration of small-scale concentration variability induced by aircraft. The photo was taken at the air field Aue/Hattorf, Germany, on 2 October 2011 at 16:15 UTC, copyright Institute of Flight Guidance, TU Braunschweig

[revised manuscript text omitted]

---

## Author Response (AR3)

**Answers to the editor on* "Studying boundary layer methane isotopy and vertical mixing processes at a rewetted peatland site by unmanned aircraft system"**

Astrid Lampert[1], Falk Pätzold[1], Magnus O. Asmussen[1], Lennart Lobitz[1], Thomas Krüger[1], Thomas Rausch[1], Torsten Sachs[1,2], Christian Wille[2], Denis Sotomayor Zakharov[3], Dominik Gaus[3], Stephan Bansmer[3], and Ellen Damm[4]

[1]TU Braunschweig, Institute of Flight Guidance, Hermann-Blenk-Str. 27, 38108 Braunschweig, Germany
[2]German Research Centre for Geosciences, Telegrafenberg, 14473 Potsdam, Germany
[3]TU Braunschweig, Institute of Fluid Mechanics, Hermann-Blenk-Str. 37, 38108 Braunschweig, Germany
[4]Alfred Wegener Institute, Helmholtz Centre for Polar and Marine Research, Am Handelshafen 12, 27570 Bremerhaven, Germany

**Correspondence:** Astrid Lampert (Astrid.Lampert@tu-braunschweig.de)

**1   Answers to the editor**

The authors would like to thank the editor for the comments on the manuscript. In the following, the comments are given in *italic*. The answers are given in normal letters. The modified text in the manuscript is given in quotation marks.

*Dear authors,*

5   *You made some improvements in the recent revised version. However, I am not satisfied with some author responses and revisions. I still consider some micrometeorological interpretation as insufficient and problematic. The text modifications partly went into the wrong direction. This necessitates some considerable improvements of the manuscript text as detailed below. The following comments mainly refer to the editor and referee comments of the previous review (with some additional comments at the end). The page and line numbers refer to the newest revised manuscript version.* We try to take into acount the comments.

10   Each point is addressed below.

*EDITOR COMMENTS 1 and 2 (of previous review)*
*a) With one (or a few) dozen of grab air samples, it is not possible to determine small-scale variabilities in the boundary layer in a meaningful and quantitative way. As mentioned in the abstract (page 1, line 20), the parallel samples should mainly be*

15   *considered as a mean to check the reliability and reproducibility of the profile measurements (see also Comment of Referee1 below). Thus, I recommend to remove the 2nd objective (page 4, lines 3-4) and the corresponding statements elsewhere in the text.*
As suggested, we removed the second objective.

*b) page 3, line 1-6: Remove this newly introduced paragraph. It is erroneous or misleading. You totally misinterpreted my original comment. The typical 30 min interval is generally used in micrometeorology to measure and calculate quantities of the boundary layer and turbulence (not just means but also variations, covariances, fluxes, spectra, etc.). This interval comprises most occurring eddy sizes and therefore prevents undersampling.*

We removed the paragraph as suggested.

*c) It still needs to be discussed whether the instantaneous point sampling is sufficient and useful for micrometeorological profile analysis.*

We changed the last sentence of Section 4.2 to: "Altogether, a clear transition in the vertical distribution of the delta values can be seen."

*d) page 3, line 4-6 (and page 12, line 8): This is an erroneous/misleading interpretation of the results of Schaller et al. (2019). They clearly concluded that the episodic 'outburst' events are not connected to soil emissions but to atmospheric dynamics (non-stationary or intermittent turbulent mixing).*

We removed the reference of the Schaller results.

*REFEREE 1 COMMENTS (of previous review)*

*e) I do not accept the response of the authors about the "different perceptions of the mixing process". From a micrometeorological perspective, mixing never amplifies concentration gradients or differences but always acts to reduce them. Thus, I want to enforce the critics of Referee 1 concerning the statements on page 12, lines 15-17. These statements should be rephrased or omitted.*

We changed the paragraph to "The differences in delta values may indicate natural inhomogeneity. However, the very dynamic circulation process around the quadrocopter has to be taken into account."

*REFEREE2 COMMENTS (of previous review)*

*f) I want to enforce Comment 2 of Referee 2 that the second objective (page 4, line 3-4) is unclear. The statement "...which require the development of new methods for understanding turbulent mixing processes" is not formulated as an objective. Considering my comment a) above, the second objective should be removed.*

As mentioned above, we removed the second objective.

*ADDITIONAL COMMENTS*

*g) page 4, line 2: Remove or rephrase the new last part of this objective ("...and therefore the spatial origin ..."). The identification of the spatial origin of the samples/layers is not provided in this study.*

We removed the second part of the sentence as suggested.

*h) page 4, line 10: Change "constant" to "near constant"*

We added "near".

*i) Fig. 3 caption: Rephrase to "Simulation of the relative vertical flow velocity (Vrel) induced by ..."* We changed as suggested.

*j) page 6, line 15: I guess this should read "3913 min$^{-1}$"*

Thank you for pointing out the error.

*k) page 11, line 24: Correct to "...except for the last ..."*

10  We corrected this.

*l) page 11, line 30: Clarify to "...that are not mixed across the inversion."*

We changed as suggested.

15  *m) page 13, line 1: What is the specific meaning/purpose of "instantaneous profiles"? Did you mean "simultaneous profiles" here?*

Yes, we corrected this.

*n) page 13, lines 5-6: Better write: "to determine vertical differences ..."*

20  We added "vertical"

*Please duly consider these comments in your further (hopefully final) revision.*

We would like to thank the editor for the clear recommendations, and hope that the new revision satisfactorily addresses the mentioned points.

[revised manuscript text omitted]

---

## Author Response (AR4)

**Answers to the editor on* "Studying boundary layer methane isotopy and vertical mixing processes at a rewetted peatland site by unmanned aircraft system"**

Astrid Lampert[1], Falk Pätzold[1], Magnus O. Asmussen[1], Lennart Lobitz[1], Thomas Krüger[1], Thomas Rausch[1], Torsten Sachs[1,2], Christian Wille[2], Denis Sotomayor Zakharov[3], Dominik Gaus[3], Stephan Bansmer[3], and Ellen Damm[4]

[1]TU Braunschweig, Institute of Flight Guidance, Hermann-Blenk-Str. 27, 38108 Braunschweig, Germany
[2]German Research Centre for Geosciences, Telegrafenberg, 14473 Potsdam, Germany
[3]TU Braunschweig, Institute of Fluid Mechanics, Hermann-Blenk-Str. 37, 38108 Braunschweig, Germany
[4]Alfred Wegener Institute, Helmholtz Centre for Polar and Marine Research, Am Handelshafen 12, 27570 Bremerhaven, Germany

**Correspondence:** Astrid Lampert (Astrid.Lampert@tu-braunschweig.de)

**1   Answers to the editor**

Dear Christof Ammann,

We thank you for your kind comments and for taking the time for the detailed comments on the manuscript. We have incorporated your advice into the revised manuscript. Below you will find our direct responses to the comments in normal letters. The comments are given in *italic*. Our changes to the text are additionally presented in quotation marks.

*Dear authors,*

*you performed the specifically requested modifications in the text. But it should be clear that other closely related parts of the text also need to be adjusted (in particular concerning my previous comments (a) and (e). Especially Section 4.2 still contains misleading and confusing statements as well as repetitions. The response to my previous comment (c) was not at all satisfying. Therefore, more improvements are necessary before publication.*

*It is not possible to give detailed suggestions for modifications in every case. However, I try to be as specific as possible in the following list of comments.*

*Christof Ammann*

*AMT Associate Editor*

Thank you again for the detailed and specific comments! We eliminated or modified statements that seem misleading.

*SPECIFIC COMMENTS*

*1. page 10, line 27/28: I suggest the following modification: "Small-scale differences in methane isotopic composition can be*

*introduced by intermittent vertical mixing processes, which may also be induced by the quadrocopter system itself."*

We changed as suggested.

5     *2. page 11, line 1-3: This is a confusing statement. The previous sentence describes the systematic vertical difference (layering) in the delta values. Such a vertical gradient is not primarily due to small-scale inhomogeneity but rather due to a significant local ground source in combination with stable stratification (i.e. a lack of vertical mixing). I suggest to modify to: "This shows that the observed systematic vertical differences are not caused by measurement uncertainty, but are due to local emissions in combination with limited mixing due to stable stratification."*

10   Thank you - your suggestion sounds indeed much better.

    *3. page 11, line 13/14: Rephrase to: "However, that latter value has been determined experimentally for long laboratory time series, while the uncertainty may be lower for subsequent analyses."*

We changed as suggested.

    *4. page 11, line 16: Omit "systematic"*

We changed as suggested.

    *5. page 11, line 18/19: This statement is a repetition of lines 11-15 and thus can be omitted.*

20   We removed all the bullet points and integrated the last one in the normal text flow.

    *6. page 11, line 20/21: The statement in the first sentence is not fully true (only for the first four flights) and, as it applies to only one set of four values, it may be purely random. The second sentence is confusing and not scientifically sound. Thus, I recommend to omit both sentences.*

25   Ok.

    *7. page 11, line 22-25: This paragraph should be integrated into the normal text flow.*

Ok.

30     *8. page 11, line 27/28: "...and the differences in delta values show a similar height profile for subsequent flights." I cannot see these 'similar height profiles' in Fig. 9 and 10. Thus, I suggest to delete this part of the sentence (or to be more specific in the explanation).*

We deleted the sentence as suggested.

35     *9. It still needs to be discussed whether the short sampling time of 1.3 s can yield representative information of the methane and isotope concentration (profile).*

*E.g. in page 11, line 30 it could be added: "However, the very dynamic circulation process around the quadrucopter and the short sampling time of only 1.3 s have to be taken into account."*

We added in the text: "However, the very dynamic circulation process around the quadrucopter and the sampling time of 1.3 s

5    have to be taken into account."

*ADDITIONAL TECHNICAL AND LANGUAGE CORRECTIONS*

*10. page 2, line 30/31: "... to determine contributions from isotopically different sources to the dominating source". This is a confusing formulation, please rephrase.*

10   We changed the text to: "Field measurements of methane isotopic composition have been performed at the Cabauw tower at a sampling altitude of 20 m, demonstrating the potential of isotopic analyses to determine contributions from isotopically different sources."

*11. page 10, line 21/22: Change to "...during stable atmospheric conditions (Wolf et al., 2017), as these prevent mixing*

15   *with..."*

We corrected the grammar.

*12. page 13, line 27: For clarification please modify to: "...the associate editor Christof Ammann..."*

We changed as suggested.

*13. page 14, line 34: Remove Denmead et al. (1998) from the reference list, as it no longer occurs in the text.*

75   We removed the reference that is not used any more.

[revised manuscript text omitted]